# The Fengyun-3D (FY-3D) global active fire product: principle, methodology and validation

**Jie Chen**[1,2,★]**, Qi Yao**[3,★]**, Ziyue Chen**[3]**, Manchun Li**[4]**, Zhaozhan Hao**[5]**, Cheng Liu**[1,2]**, Wei Zheng**[1,2]**, Miaoqing Xu**[3]**, Xiao Chen**[3]**, Jing Yang**[3]**, Qiancheng Lv**[3]**, and Bingbo Gao**[5]

[1]Innovation Center for FengYun Meteorological Satellite, National Satellite Meteorological Center (National Center for Space Weather), China Meteorological Administration, Beijing 100081, China
[2]Key Laboratory of Radiometric Calibration and Validation for Environmental Satellites, National Satellite Meteorological Center (National Center for Space Weather), China Meteorological Administration, Beijing 100081, China
[3]College of Global Change and Earth System Science, Beijing Normal University, Beijing 100091, China
[4]School of Geographic and Oceanographic Sciences, Nanjing University, Nanjing 210008, China CE1
[5]College of Land Science and Technology, China Agricultural University, Beijing 100083, China
★These authors contributed equally to this work.

**Correspondence:** Ziyue Chen (zychen@bnu.edu.cn) and Bingbo Gao (gaobingbo@cau.edu.cn)

**Abstract.** TS1 CE2 Wildfires have a strong negative effect on the environment, ecology and public health. However, the potential degradation of mainstream global fire products leads to large uncertainty in the effective monitoring of wildfires and their influence. To fill this gap, we produced Fengyun-3D (FY-3D) global fire products with a similar spatial and temporal resolution, aiming to serve as an alternative to and continuity for Moderate Resolution Imaging Spectroradiometer (MODIS) global fire products. Firstly, the sensor parameters and major algorithms for noise detection and fire identification in FY-3D products were introduced. For visual-check-based accuracy assessment, five typical CE3 regions across the globe, Africa, South America, the Indochinese Peninsula, Siberia and Australia, were selected, and the overall accuracy exceeded 94 %. Meanwhile, the consistence between FY-3D and MODIS fire products was examined. The result suggested that the overall consistence was 84.4 %, with a fluctuation across seasons, surface types and regions. The high accuracy and consistence with MODIS products proved that the FY-3D fire product is an ideal tool for global fire monitoring. Based on field-collected reference data, we further evaluated the suitability of FY-3D fire products in China. The overall accuracy and accuracy without considering omission errors were 79.43 % and 88.50 % higher, respectively, than those of MODIS fire products. Since detailed local geographical conditions were specifically considered, FY-3D products should be preferably employed for fire monitoring in China. The FY-3D fire dataset can be downloaded at http://satellite.nsmc.org.cn/portalsite/default.aspx (NSMC, 2021).

## 1 Introduction

More than half of global land surfaces have been influenced by wildfires, and the total global burned area adds up to the area of the European Union every year (Andela et al., 2019; Keeley et al., 2011; Moritz et al., 2012). Wildfires, especially large-scale wildfires, in forests, grasslands and farmlands, have a significant impact on crop productivity (Jethva et al., 2019), atmospheric pollution (Guo et al., 2020), biodiversity (Kelly et al., 2020), climate change (Alisjahbana et al., 2017 TS2; Keegan et al., 2014) and public health (Huff et al., 2015; Johnston et al., 2012; Oliveira et al., 2020; Yuchi et al., 2016). In recent years, the increasing events of forest fires in China, the USA, Australia, and Amazon rain forests

and grassland fires in Mongolia have caused a large number of casualties (Cochrane, 2003), the loss of millions of wild animals (Wintle et al., 2020), remarkably deteriorated air quality (Guo et al., 2010; Liu et al., 2018; Marlier et al., 2012; Volkova et al., 2019), severely damaged ecosystems (Cerda et al., 2012), massive economic losses (Stephenson et al., 2013), and regional or global climate change (Abram et al., 2021; Jacobson, 2014; Twohy et al., 2021; Wang et al., 2020).

Due to wildfires' great influences, growing emphasis has been placed on the monitoring of wildfires based on remote sensing products. Since the 1970s, the implementation of and research into satellite-based fire detection have been in the USA using National Oceanic and Atmospheric Administration (NOAA) series satellites (http://www.noaa.gov TS3; Dozier et al., 1981 TS4; Flannigan and Haar, 1986; Kaufman et al., 1990; Boles et al., 2000 TS5). NOAA fire products, with a spatial resolution of 1.1 km and a daily temporal resolution, have been employed globally for decades and provide data support for long-time-series analysis. In addition to NOAA fire products, a diversity of regional or global fire products have been proposed in recent years.

Thanks to its easy access, long time series and reliable accuracy (Giglio et al., 2018), the Moderate Resolution Imaging Spectroradiometer (MODIS) fire product, with a spatial resolution of 1 km and a temporal resolution of 12 h and available since 2000, has become one of the most widely employed fire products to monitor the temporal evolution of large-scale wildfires, including forest fires (Mohajane et al., 2021), grassland fires (Zhang et al., 2017) and crop residue burning (Li et al., 2016). With a similar temporal resolution (12 h), the Visible Infrared Imaging Radiometer Suite (VIIRS) fire products with a spatial resolution of 375 m have been available for fire detection since 2011. Despite a higher spatial resolution, VIIRS fire products are produced using fewer bands than MODIS fire products, and the mainly used 4 μm I-band may lead to large bias in the estimation of FRP (fire radiative power) during an intense fire event (Schroeder et al., 2014). Consequently, VIIRS fire products present a relatively poor consistence with MODIS fire products and the accuracy of VIIRS fire products is generally lower than that of MODIS fire products (Sharma et al., 2017). In this regard, VIIRS fire products may not serve as a complete replacement and should be comprehensively employed with MODIS fire products.

In recent years, with the growing need for real-time monitoring of a diversity of environmental issues and ecological processes, some satellites have been launched to provide remote sensing products with extremely high temporal resolution. GEOS-16 Advanced Baseline Imager (ABI) active fire products, with a temporal resolution of 5 min and a spatial resolution of 2 km, have been available since 2017 (Hall et al., 2019). GEOS-ABI fire products can effectively monitor medium- to large-scale fires and be used for estimating fire emissions. GEOS-ABI fire products may lead to a poor detection accuracy when identifying small-scale fires (Li et al., 2020). GEOS-ABI mainly provides regional fire products in the southeastern conterminous United States (CONUS). Himawari-8 products, with a spatial resolution of 2 km and temporal resolution of 10 min, have been widely employed to monitor meteorology and wildfires in Asia and Australia since 2015 (Xu et al., 2017). Similarly to GEOS-16 ABI fire products, Himawari-8 fire products are also limited in effectively detecting small-scale fires (Wickramasinghe et al., 2018). Despite an extremely high temporal resolution, fire products produced using geostationary satellites only cover a regional area and cannot monitor the distribution and evolution of wildfires at a global scale.

Long-term running leads to the aging of sensors (Sayer et al., 2015; Liu et al., 2017; Barnes et al., 2019) and causes the degradation of sensor sensitivities (Lyapustin et al., 2014; Doelling et al., 2015; Xiong et al., 2019), increased system errors (Fensholt et al., 2012 TS6; Xie et al., 2011) and decreased product quality (Fang et al., 2012; Wang et al., 2012). With a high temporal resolution and so far the longest time series, MODIS global fire products have become the most important data source for examining historical regional and global fires, monitoring occurring fires, and investigating their environmental influences. However, after 22 years of running, the gradual aging of sensors will cause, if it has not already, the degradation of MODIS global fire products. To continuously make full use of the existing long-term series of MODIS fire products, even if MODIS degrades or stops services in the future, a fire product with good reliability, good consistence and similar characteristics is urgently needed to serve as a potential alternative to and continuity for global MODIS fire products. Since the launch of the Fengyun-3C (FY-3C) satellite in September 2013, a series of FY meteorological satellites have been designed to produce global active fire products. FY-3C VIRR CE4 fire products were produced based on an effective active fire detection algorithm (Lin et al., 2017), which considered dynamic thresholds and infrared gradients. However, the overall accuracy of FY-3C VIRR fire products remained unsatisfactory at the global scale and have thus not been publicly released.

In November 2017, the Fengyun-3D (FY-3D) satellite was launched with an improved Medium Resolution Spectral Imager (MERSI) for fire detection. With a similar spatiotemporal resolution, FY-3D provides a promising solution for the continuity of global MODIS fire products. In this paper, we introduce the characteristics and fire detection algorithms of a new global fire product based on FY-3D (recently downloadable from our official website http://satellite.nsmc.org.cn/portalsite/default.aspx TS7). Through visual check, consistence check and accuracy assessment based on ground truth data, the FY-3D global fire product is comprehensively compared with the MODIS global fire product at the global and regional scale. Thanks to its good global consistence and regional suitability, the FY-3D global fire product has the potential to serve as a continuity of the global MODIS fire prod-

uct and better support ecological and environment research in China.

## 2 Overview of FY-3 fire products

### 2.1 Instrument

As one of the core instruments of the Fengyun-3 (FY-3) satellites, the updated Medium Resolution Spectral Imager (MERSI) has become one of the most advanced remote sensing instruments based on wide-swath imaging. The FY-3D satellite was launched in November 2017 with 10 sets of remote sensing instruments, including the Medium Resolution Spectral Imager II (MERSI-II). MERSI-II integrates the functions of the two original imaging instruments (MERSI-I and VIRR) of FY-3B and FY-3C, with a total of 25 channels, including visible light, near infrared, medium infrared and far infrared (as in Table 1). The infrared imaging, detection sensitivity and calibration accuracy of MERSI-II are improved greatly. It is the first imaging instrument that can access the 250 m resolution infrared split-window area globally and capture seamless 250 m resolution true-color global images on a daily basis. MERSI-II also enables the high-quality retrieval of atmospheric, land and marine parameters such as clouds, aerosols, vapor, land surface features and ocean color, supporting global support for environment and climate issues.

### 2.2 Product overview

There are two middle-infrared bands (3.8 and 4.05 μm TS9 ), and both are sensitive to strong heat signals. Their differences lie in their performance under different temperature and radiation conditions. The 3.8 μm band is closer to the wavelength of solar radiation and has better reflection under solar radiation. As a comparison, the 4.05 μm band more easily misses weak fires. Therefore, current FY-3D fire products are mainly produced based on the 3.8 μm band for better fire identification. According to the calculation, the emissivity of forest and grassland fires in the mid-infrared band can be hundreds of times higher than that of the surface at normal temperature, making the radiance and brightness temperature of the fire spot significantly higher than surrounding pixels. For rapid monitoring of global wildfires, it is necessary to develop an algorithm for the automatic identification of fire spots.

MERSI-II fire monitoring products from the FY-3D satellite can provide fire spot location, sub-pixel fire spot area, temperature and fire spot intensity in inland areas around the world and generate global fire spot pixel information (including day and night) in HDF files. FY-3D fire products are produced following a projection with equal latitude and longitude (0.01°). Fire spot intensity is classified according to the sub-pixel fire spot area and temperature, with an overall accuracy above 85 %. Based on daily monitoring products,

the SMART (Satellite Monitoring Analyzing and Remote sensing Tools CE5 ) system can generate the images of global monthly fire spot distribution, with a resolution of 0.25°.

The algorithm for fire spot identification depends on the sensitivity of mid-infrared channels to high-temperature heat sources. The radiance and brightness temperature of the pixels in the mid-infrared channels with sub-pixel fire spots are higher than those of the surrounding non-fire pixels and those of the pixels in the far-infrared channels. Therefore, the pixels with fire spots can be identified by setting an appropriate threshold, and the estimation of background temperature is the key to high detection accuracy and sensitivity.

Sub-pixel fire spot estimation relies on the brightness temperature in mid-infrared channels, and the far-infrared channels are employed when the mid-infrared channels have saturated brightness temperature. In the single-channel estimation formula, the temperature of the open-flame spot is set to 750 K.

Fire spot intensity, namely fire radiation power (FRP), is obtained by substituting the area and temperature of sub-pixel fire spots into the Stefan–Boltzmann formula of full-band blackbody radiation.

$$J^* = \varepsilon \sigma T^4 \tag{1}$$

The radiant emittance $J^*$ has dimensions of energy flux, and the SI units of measure are joules per second per square meter. The SI unit for absolute temperature $T$ is the kelvin. $\varepsilon$ is the emissivity for the grey body; if it is a blackbody, $\varepsilon = 1$. $\sigma$ TS10 is the Stefan–Boltzmann constant.

FRP is divided into 10 levels, indicating different ranges of radiation intensity and the fire behavior at fire spot pixels. Fire spots are classified into four groups with regard to credibility, namely the real fire spots, possible fire spots, fire spots affected by the cloud and noisy fire spots (disturbed by clouds and noise).

FY-3D MERSI-II daily global fire monitoring products are illustrated in Fig. 1. The major processing of daily fire spot products is the generation of 5 min fire spot lists, which include such information as the observation time of fire spot pixels, latitude and longitude, the sub-pixel fire spot area and temperature, and FRP. Next, all the 5 min fire spot information for each day is merged into the daily global fire information list.

FY-3D MERSI-II monthly global fire monitoring products consist of the information list of global fire spot pixels and the density map of global fire spots. The information list of monthly global fire spots covers all global fire spot pixels in the particular month. Concerning the multi-time monitoring information of the same pixel, the maximum fire spot area is taken as the current-month fire spot information for the pixel. Figure 2 is an illustration of the density map of global fire spots based on FY-3D MERSI-II, in which different colors indicate the number of fire spot pixels on a 0.25° × 0.25° spatial grid. Compared with daily FY-3D fire products, monthly FY-3D fire products were advantageous

**Table 1.** Major channel parameters of FY-3D MERSI-II (compared with MODIS/Aqua). TS8

| Channel | | Wavelength (μm) | | Waveband | | Resolution (km) | | Application |
|---|---|---|---|---|---|---|---|---|
| MERSI | MODIS | MERSI | MODIS | MERSI | MODIS | MERSI | MODIS | |
| 1 | 3 | 0.470 | 0.469 | Visible light | | 0.25 | 0.50 | Ocean color, land |
| 2 | 4 | 0.550 | 0.555 | Visible light | | 0.25 | 0.50 | |
| 3 | 1 | 0.650 | 0.645 | Visible light | | 0.25 | 0.25 | Land, cloud |
| 4 | 2 | 0.865 | 0.859 | Near infrared | | 0.25 | 0.25 | Ocean color, vegetation |
| 5 | 5 | 1.380 | 1.380 | Near infrared | | 1.00 | 0.50 | Land, cloud, snow |
| 6 | 6 | 1.640 | 1.640 | Near infrared | | 1.00 | 0.50 | |
| 7 | 7 | 2.130 | 2.130 | Near infrared | | 1.00 | 0.50 | Land, cloud |
| 8 | 8 | 0.412 | 0.412 | Visible light | | 1.00 | 1.00 | Ocean color, phytoplankton, biogeochemistry |
| 9 | 9 | 0.443 | 0.443 | Visible light | | 1.00 | 1.00 | |
| 10 | 10 | 0.490 | 0.488 | Visible light | | 1.00 | 1.00 | |
| 11 | 12 | 0.555 | 0.555 | Visible light | | 1.00 | 1.00 | |
| 12 | 13 | 0.670 | 0.667 | Visible light | | 1.00 | 1.00 | |
| 13 | – | 0.709 | – | Visible light | | 1.00 | – | |
| 14 | 15 | 0.746 | 0.748 | Visible light | | 1.00 | 1.00 | |
| 15 | 16 | 0.865 | 0.869 | Near infrared | | 1.00 | 1.00 | |
| 16 | 17 | 0.905 | 0.905 | Near infrared | | 1.00 | 1.00 | Atmosphere, water vapor |
| 17 | 18 | 0.936 | 0.936 | Near infrared | | 1.00 | 1.00 | |
| 18 | 19 | 0.940 | 0.940 | Near infrared | | 1.00 | 1.00 | |
| 19 | 26 | 1.040 | 1.040 | Near infrared | | 1.00 | 1.00 | Cirrus |
| 20 | 20 | 3.800 | 3.750 | Medium infrared | | 1.00 | 1.00 | Surface, cloud, atmospheric temperature |
| 21 | 23 | 4.050 | 4.050 | Medium infrared | | 1.00 | 1.00 | |
| 22 | 28 | 7.200 | 7.325 | Far infrared | | 1.00 | 1.00 | Water vapor |
| 23 | 29 | 8.550 | 8.550 | Far infrared | | 1.00 | 1.00 | |
| 24 | 31 | 10.800 | 11.030 | Far infrared | | 0.25 | 1.00 | Surface, cloud temperature |
| 25 | 32 | 12.000 | 12.020 | Far infrared | | 0.25 | 1.00 | |

in revealing the global patterns of fire spots. As shown in Fig. 2, the global fire spots were mainly distributed in southern Africa, central South America, southern North America, north-central Asia and northern Australia in June 2019.

## 3 Methods

This section mainly introduces the specific algorithm and steps for generating FY-3D global fire products based on the original data obtained from MERSI-II. The input data include MERSI-II global orbital Earth observations, MERSI-II global orbital geographical locations, MERSI-II global orbital cloud detection data, and global land and sea template data, as shown in Table 2.

Automatic identification of fire spots is the major step for generating fire products. Firstly, the 5 min level-1 (L1) data segments of MERSI-II and various auxiliary data are read in, and the noise lines are identified to generate the noise line mark. Next, the 5 min data segments are projected according to the rule of the equal latitude and longitude and cut as $5° \times 5°$ grids to generate a local map.

Secondly, fire spots in each $5° \times 5°$ local map are identified pixel by pixel, subject to the calculation of the sub-pixel fire spot area and the estimation of FRP. According to their credibility, the identified fire spot pixels are classified into four categories. Subsequently, all the $5° \times 5°$ local fire spot information in the 5 min data segments is synthesized to generate fire spot HDF file products. The general steps for producing FY-3D fire products are briefly shown in Fig. 3, and the detailed procedures are explained as follows.

### 3.1 The general principle of fire detection based on MERSI-II

Channel 20 of FY-3D MERSI-II is mid-infrared, with a wavelength of 3.55–3.95 μm, while Channels 24 and 25 are far-infrared, with a wavelength of 10.3–11.3 and 11.5–12.5 μm, respectively. According to Wien's displacement law,

$$\lambda \cdot T = b, \tag{2}$$

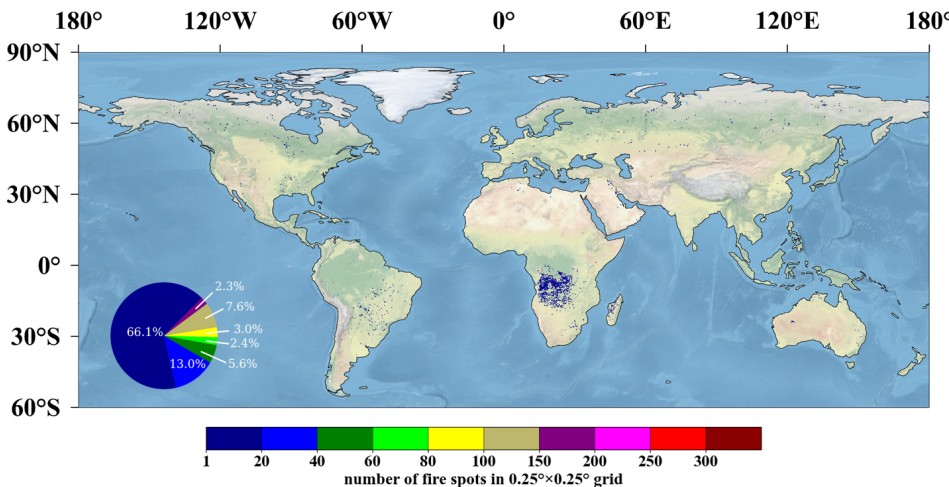

**Figure 1.** Thematic map of global fire monitoring by FY-3D (13 June 2019). The color bar with different colors means the number of fire spots in the 0.25° × 0.25° grid.

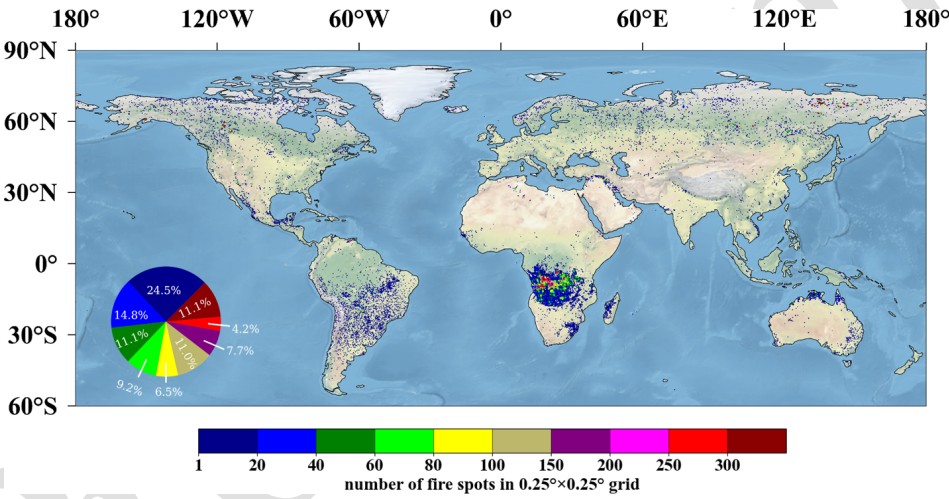

**Figure 2.** Density map of global fire spots based on FY-3D (June 2019). Fire-prone areas were distributed in northern Russia, south-central Africa, southeastern South America, the coastland of Australia and small parts of Canada.

where λ is the peaks at the wavelength; $T$ is the absolute temperature; and $b$ is a constant of proportionality called Wien's displacement constant, equal to about 2898 μm K. Blackbody temperature $T$ is inversely proportional to peak radiation wavelength $\lambda_{max}$, as the higher temperature can lead to the smaller peak radiation wavelength. The peak radiation wavelength of the surface at normal temperature (about 300 K) is close to that of Channels 24 and 25; the combustion temperature of forest fires is generally 500–1200 K, and the peak wavelength of thermal radiation is close to that of Channel 20. When a fire spot appears in the observed pixel, the radiance increment in Channel 20 caused by the high temperature in the small sub-region of the pixel, where the fire spot is located (since the pixel resolution of the scanning radiometer is 1.1 km, CE7 it is usually not be all open-flame areas at the same time in such a large range), is much higher than sur-

rounding pixels without an open flame and also greater than that in Channels 24 and 25. In this case, the weighted averages of the radiance increase and brightness temperature increase of each channel differ notably in this pixel, based on which the fire information can be extracted and analyzed.

As indicated by Fig. 4a, when the fire spot temperature grows, the brightness temperature of Channel 20 (CH20) pixels increases rapidly. Even if the fire spot only accounts for 0.1 % the pixel area, the brightness temperature increment can reach 10 K (44 K) when the fire spot is 500 K (900 K). Although the brightness temperature increase of CH24 also rises with the higher fire spot temperature, it is far lower than that of CH20. Figure 4b illustrates that as the fire spot area becomes larger, the brightness temperature of CH20 mixed pixels grows rapidly. It reaches 12 K when the fire spot is 900 K, even if the fire spot only accounts for 0.01 % of the

**Table 2.** Input file list of MERSI-II global fire monitoring software. CE6

| No. | Item | Format | Data type | Period | Source | Description |
|---|---|---|---|---|---|---|
| 1 | MERSI-II global orbital Earth observations | HDF | 1B | Real time | Preprocessor | Data file after preprocessing 5 min data segments of MERSI-II |
| 2 | MERSI-II global orbital geolocations | HDF | Float | Real time | Preprocessor | Locations after preprocessing 5 min data segments of MERSI-II |
| 3 | MERSI-II global orbital cloud detection data | HDF | Float | Real time | Product system | 5 min cloud detection products of MERSI-II produced by the product system |
| 4 | Global land and sea template data | DAT | Grid | Static | Data management and user service subsystem | Global land–sea boundaries |

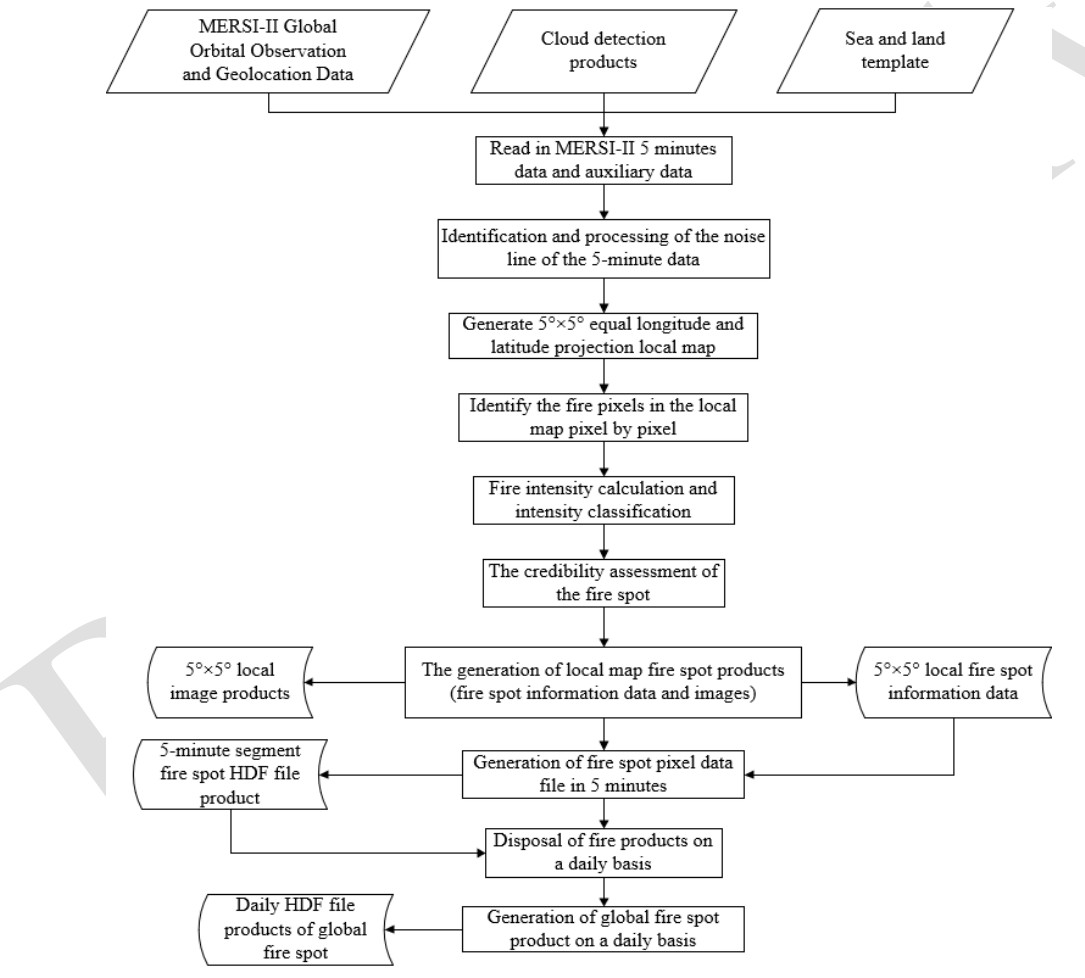

**Figure 3.** General flowchart for generating FY-3D MERSI-II fire spot products.

pixel area. Similarly, the brightness temperature increment of CH24 grows at a much lower rate than that of CH20.

## 3.2 Automatic identification algorithms for fire spots

### 3.2.1 Detection of cloud pixels

Effective cloud detection is required for generating reliable fire products for the following reasons. Firstly, the existence of cloud in the atmospheric layers may block the emitted in-

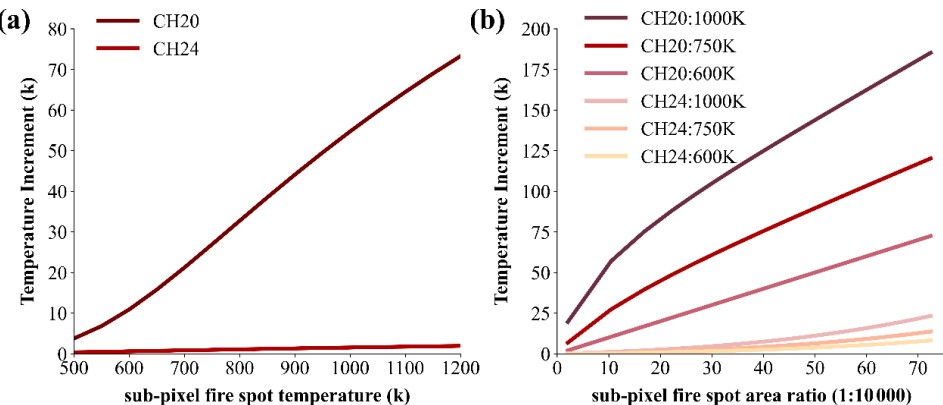

**Figure 4. (a)** Curves of FY-3D MERSI-II CH20 and CH24 brightness temperature increment with fire spot temperature (with fire spot area accounting for 0.1 % of pixel area and background temperature at 290 K). **(b)** Curves of FY-3D MERSI-II CH20 and CH24 brightness temperature increment with fire spot area (with fire spot temperature at 600, 750 and 1000 K; background temperature at 290 K; and the ratio of fire spot area to pixel area increasing from 0.01 % to 0.4 %).

formation of fire spots, leading to missed identification. Secondly, specular reflection of cloud can lead to wrong identification of fire spots. Therefore, cloud identification was conducted before fire identification. Similarly to MODIS, FY-3D also included radiation information from multiple bands, and the principle of cloud identification for FY-3D fire products was similar to that of MODIS. Based on the reflectance difference between cloud and land pixels, we classified cloud pixels following the rules listed in Table 3.

### 3.2.2 Calculation of background temperature

According to the principle of fire spot identification, when a fire spot appears in a pixel (i.e., open flame), the brightness temperature of the pixel in Channel 20 is significantly higher than the background brightness temperature (the brightness temperature of surrounding non-fire pixels); the brightness temperatures of Channels 24 and 25 are also higher than the background, but the temperature difference is much smaller than that of Channel 20. In this case, the difference in brightness temperature between fire spot pixels and background in both the mid-infrared channel and far-infrared channels can be employed as important factors for automatic identification of fire spots. Therefore, the background temperature of the detected pixel is required for identifying fire spots. Since the background temperature cannot be obtained from the fire spot pixels, it should be calculated according to the average of their surrounding pixels. However, the reflection of solar radiation during the daytime also causes a higher brightness temperature in the mid-infrared channel, which mainly occurs in the zone bare of vegetation, cloud surface and water bodies (specular reflection). In particular, the difference in brightness temperature between mid-infrared and far-infrared channels caused by specular reflection of solar radiation can reach tens of kelvins on the cloud surface and water bodies. Since the reflection of solar radiation on the bare surface is relatively weak in the mid-infrared channel, a few degrees of difference can cause non-fire pixels misclassified as fire pixels due to the high sensitivity requirement for fire identification. When the background brightness temperature is calculated, pixels that already contain fire spots should also be excluded. Therefore, suspected high-temperature pixels, which may already contain fire spot pixels, cloudy pixels, water pixels and those pixels affected by solar flare, should be removed for background temperature calculation.

Furthermore, the pixel size in the mid-infrared channel of a meteorological satellite is about $1\,\mathrm{km}^2$. Within this range, the underlying surface may be diversified and composed of sub-regions with different fractional vegetation cover (FVC). In the daytime, affected by solar radiation, the brightness temperature of different FVC types may vary, making the calculated background temperature higher than expected. To address this issue, Kaufman et al. (1998) suggested the use of the standard deviation of background temperature for fire identification, which significantly reduced the overestimation of background temperature caused by different underlying surfaces.

After the abovementioned disturbing pixels were removed, the average and standard deviation of background temperature in the mid-infrared channel and the background average and standard deviation of brightness temperature difference between the mid-infrared and far-infrared channels were calculated with peripheral pixels as background pixels.

The calculation of background temperature was acquired in the following steps. For each $3 \times 3$ window, the background temperature is calculated as the mean temperature of all background pixels. Suspicious high-temperature pixels can be identified according to the following conditions:

$$T_{\mathrm{Mir}} > T_{\mathrm{th}} \text{ or } T_{\mathrm{Mir}} > T'_{\mathrm{Mir\_bg}} + T_{\mathrm{Mir\_bg}} ,$$

where $T_{\mathrm{Mir}}$ is the bright temperature CE9 in the middle-infrared channel. $T_{\mathrm{th}}$ is the threshold for high-temperature

**Table 3.** Major rules for cloud pixel identification.

| Number | Conditions |
|---|---|
| 1 | $T_{\mathrm{Mir}} - T_{\mathrm{far1}} < 4\,\mathrm{K}$ |
| 2 | $T_{\mathrm{Mir}} - T_{\mathrm{far1}} > 20\,\mathrm{K}$ and $T_{\mathrm{Mir}} < 285\,\mathrm{K}\ |T_{\mathrm{far1}}| < 280\,\mathrm{K}$ |
| 3 | $R_{\mathrm{Vis}} > 0.28$ and SolarZenith $< 70°\ |$ SolarZenith $< 60°$ and SateZenith $< 60°$ |
| 4 | $T_{\mathrm{far1}} < 265\,\mathrm{K}$ |
| 5 | $T_{\mathrm{Mir}} < 270\,\mathrm{K}$ and $T_{\mathrm{far1}} - T_{\mathrm{far2}} < 4\,\mathrm{K}$ |
| 6 | $T_{\mathrm{far1}} < 270\,\mathrm{K}$ and $T_{\mathrm{far1}} - T_{\mathrm{far2}} > 60\,\mathrm{K}$ |
| 7 | $T_{\mathrm{Mir}} < 320\,\mathrm{K}$ and $T_{\mathrm{Mir}} < T_{\mathrm{Mir\_TH}}$ |
| 8 | SolarZenith $> 70$ and $R_{\mathrm{Vis}} > 0.28\,T_{\mathrm{Mir}} < 320\,\mathrm{K}$ |

$T_{\mathrm{Mir}}$: mid-infrared channel; $T_{\mathrm{far1}}$: 10.8 μm far-infrared channel; $T_{\mathrm{far2}}$: 12 μm far-infrared channel; $R_{\mathrm{Vis}}$: visible light channel; SolarZenith: solar zenith angle; SateZenith: satellite zenith angle. Note that these eight rules are set to exclude a diversity of cloud biases. CE8 And a pixel that meets any rule in Table. TS11

pixels in the middle-infrared channel, usually set as the sum of the mean bright temperature of all pixels in the window and $2\times$ its corresponding standard deviation. $T'_{\mathrm{Mir\_bg}}$ is the mean bright temperature of background pixels.

$T_{\mathrm{Mir\_bg}}$ is the allowed difference between the mean background bright temperature and the suspicious high-temperature pixel, usually set as $2.5\times$ the standard deviation of background pixels. If there were less than 20 % of pixels were cloudless pixels, then the $3 \times 3$ window was extended to $5 \times 5, 7 \times 7, 9 \times 9, \ldots, 51 \times 51$. If still not applicable, then this pixel was marked as a non-fire pixel.

### 3.2.3 Identification of fire pixels

With obtained background temperature, the difference between brightness temperature and background temperature in the mid-infrared channel, as well as the difference in brightness temperature and background temperature between mid-infrared and far-infrared channels, at the candidate pixels could be calculated, based on which we could decide whether the threshold of fire spot identification was reached. If the threshold was reached, the pixel is preliminarily marked as a fire pixel. Next, for daytime observation data, it is necessary to further check whether the increase in brightness temperature in the mid-infrared channel was interfered with by solar radiation in the cloud area. Through the two-stage check, fire pixels could be effectively extracted.

When the following two conditions are met, a pixel can be identified as a fire pixel:

- $T_{3.9} > T_{3.9\mathrm{bg}} + n_1 \times \delta T_{3.9\mathrm{bg}}$ .

- $T_{3.9\_11} > T_{3.9\mathrm{bg\_11bg}} + n_2 \times \delta T_{3.9\mathrm{bg\_11bg}}$ .

Here $T_{3.9}$ is the bright temperature of the pixel at 3.9 μm. $T_{3.9\mathrm{bg}}$ is the background bright temperature. $\delta T_{3.9\mathrm{bg}}$ is the standard deviation of the bright temperature of background pixels. $T_{3.9\_11}$ is the difference in bright temperature between 3.9 and 11 μm. $T_{3.9\mathrm{bg\_11bg}}$ is the difference in background bright temperature between 3.9 and 11 μm. The setting of this condition aimed to identify the difference in land

cover types in the window. When the land cover types in the window were generally consistent, $\delta T_{3.9\mathrm{bg\_11bg}}$ is relatively small. For the identification of fire pixels, when $\delta T_{3.9\mathrm{bg\_11bg}}$ was smaller than 2 k CE10, this value was replaced using 2 K. When $\delta T_{3.9\mathrm{bg\_11bg}}$ was larger than 4 k, this value was replaced using 4 K. $n_1$ and $n_2$ are background coefficients, which vary across regions, observation time and observation angles. For instance, for northern grasslands, $n_1$ and $n_2$ were set as 3 and 3.5, respectively.

### 3.2.4 Identification of noise line

Satellite data received by the ground system contain noise. For instance, some scanning lines may contain many noisy pixels that affect fire spot identification. In this case, noise lines, referred to multiple consecutive noisy pixels in one scanning line, should firstly be checked. Since the identification of fire was carried out on the areal map projected with an equal latitude and on the same circle of longitude, the identified latitude and longitude of fire spots failed to reflect the original positions of scanning lines. Therefore, the noise line was identified on the 5 min data segments before projection. Firstly, the 5 min data segments were employed to identify fire spots, and the line number of identified fire spot pixels was recorded. Following this, the number of fire spot pixels in each line was counted. When the number of fire spot pixels in a line exceeded the empirical threshold, it was identified as a noise line and all pixels in the line are marked as noisy ones. In the following process, all pixels in this line were no longer considered for fire spot identification.

### 3.3 Estimation of fire radiation power (FRP)

FRP can be calculated using the Stefan–Boltzmann formula (Matson et al., 1984) TS12 through the estimation of the sub-pixel fire spot area and temperature.

### 3.3.1 Estimation of sub-pixel fire spot area and temperature

MERSI-II data are 12 bit, with a quantization level of 0–4095 and high radiation resolution. The spatial resolution is 1.1 km, and the radiance of a pixel observed by the satellite is the weighted average of the radiance of all the ground objects within the pixel range as

$$N_t = \left( \sum_{i=1}^{n} \Delta S_i N_{T_i} \right) \Big/ S, \tag{3}$$

where $N_t$ is the radiance of the pixel observed by the satellite, $t$ is the brightness temperature corresponding to $N_t$, $S_i$ is the area of the $i$th sub-pixel, $N_{T_i}$ is the radiance of the sub-pixel, $T_i$ TS13 is the temperature of the sub-pixel and $S$ is the total area of the pixel.

Due to different FRP levels and temperatures, underlying surfaces containing fire spots can be divided into fire zones and non-fire zones (background). When fire spots appear, the radiance of pixels containing fire spots (i.e., mixed pixels) can be expressed by the following formula:

$$N_{i\text{mix}} = P \cdot N_{i\text{hi}} + (1-P) \cdot N_{i\text{bg}}$$
$$= P \cdot \frac{C_1 V_i^3}{e^{\frac{C_2 V_i}{T_{\text{hi}}}} - 1} + (1-P) \cdot \frac{C_1 V_i^3}{e^{\frac{C_2 V_i}{T_{\text{bg}}}} - 1}, \tag{4}$$

where $P$ is the percentage of the sub-pixel fire spot area in the pixel; $N_{i\text{mix}}$, $N_{i\text{hi}}$ and $N_{i\text{bg}}$ are the radiance of mixed pixels, the sub-pixel fire spot (fire zone) and surrounding background; $T_{\text{hi}}$ and $T_{\text{bg}}$ are the temperature of sub-pixel fire spots and background; $V_i$ is the central wavenumber of channels; and $C_1$ and $C_2$ are Planck constants.

For Eq. (4), there are two unknown variables, $P$ and $T_{\text{hi}}$. According to the characteristics of infrared channels in the scanning radiometer (dynamic brightness temperature and spatial resolution), the radiation increase in high-temperature sources varies notably in different bands. To address this issue, a strategy is employed to estimate the actual area and temperature of fire spots according to the radiation in different infrared channels. When the mid-infrared channel was not saturated, it was used for estimating the sub-pixel fire spot area and temperature. Otherwise, the far-infrared channel was alternatively employed for estimation.

When a single channel was adopted to estimate the sub-pixel fire spot area, the fire spot temperature was set to an appropriate value, which was 750 K in this product.

### 3.3.2 Calculation of fire radiation power

Based on the percentage of the sub-pixel fire spot area, $P$, and fire spot temperature, FRP can be calculated using the Stefan–Boltzmann formula:

$$\text{FRP} = P \cdot S_{\lambda,\varphi} \cdot \sigma T^4, \tag{5}$$

where FRP is fire radiation power (W); $S_{\lambda,\varphi}$ is the sub-pixel fire spot area of pixels located at longitude $\lambda$ and latitude $\varphi$, which is calculated according to the percentage of the sub-pixel fire spot area $P$ and the total pixel area; $T$ is the sub-pixel fire spot temperature and set to 750 K; and $\sigma$ is the Stefan–Boltzmann constant, $5.6704 \times 10^{-8}$ (W m$^{-2}$ K$^{-4}$).

### 3.4 Verification methods

Wildfires are characterized by random and rapid changes, so it is difficult to verify the product accuracy of GFR (global fire) according to actual ground information. In this paper, the accuracy of FY-3 fire products is tested through visual interpretation and cross-verification of other products. Specifically, due to the extremely large size of GFR datasets, we set the different strategies for accuracy assessment. For visual interpretation, several 5 min data segments with regional representation were selected for verification using manually identified fire spots. For cross-verification with other fire products, global fire spot data throughout 2019 were employed.

The error was defined as the distance from the positions (longitude and latitude) of automatically identified fire spot pixels to corresponding manually identified ones. When the difference in latitude and longitude was less than or equal to 0.02°, the automatically identified pixel was regarded as a successful identification.

$$\sqrt{(\text{lat1} - \text{lat2})^2 + (\text{long1} - \text{long2})^2} \leq 0.02°, \tag{6}$$

where lat1 and lat2 are the latitude of PGS (Product Generation System CE11) fire spot pixels and manually identified pixels (reference pixels) and long1 and long2 are the longitude of PGS fire spot pixels and manually identified pixels (reference pixels), respectively.

In addition to the visual-check-based accuracy assessment at the global scale, we also employed a set of field-collected reference data to verify the suitability of FY-3D in China, which is further explained in the following sections.

## 4 Results

### 4.1 Global accuracy assessment of FY-3D fire products based on visual interpretation

In this research, 5 min segments of FY-3D fire products in different continents, including Africa, South America, the Indochinese Peninsula, Siberia and Australia, were collected at 12:15 (UTC) on 13 June 2018, 17:05 (UTC) on 21 August 2019, 06:15 (UTC) on 13 March 2019, 03:40 (UTC) on 13 November 2019 and 17:40 (UTC) on 29 May 2018, respectively, for visual interpretation. The specific observation positions are shown in Fig. 5 with five corresponding fire detection pictures of FY-3D.

These regions were selected for evaluating the global reliability of FY-3D fire products for the following reasons. Firstly, Africa, South America, the Indochinese Peninsula, Siberia and Australia are the regions with the most frequent

fire events across the globe. Secondly, there is rich vegetation in these regions, which provides the foundation for stable combustion across a year. Thirdly, these regions cover large areas with generally unified underlying surfaces. Fourthly, these areas are of regional representation: Siberia represents typical regions with frequent forest fires in the Northern Hemisphere. Africa represents typical tropical grasslands and forests in the Equator regions. South America represents virgin tropical rain forests.

Figure 5 presents the spatial distribution of GFR spots and manually identified fire pixels in the 5 min segment of the above regions. According to Fig. 5b, most fire spots in FY-3D products and manually extracted fire spots in South America were in the same positions. In Fig. 5c, most FY-3D and manually extracted fire spots in Africa coincided or were in a close position. In Fig. 5d, despite a few mismatched fire spots, the positions of FY-3D and manually extracted fire spots in the Indochinese Peninsula were consistent. Figure 5e and f also show that most fire spots are matched in Russia and Australia. Table 4 shows accuracy of GFR spots in the five typical regions. The accuracy of automatically identified fire spot in all regions was generally consistent and all exceeded 90 %. Since these selected regions represented distinct vegetation types and are located in different hemispheres, the verification of FY-3D fire products based on 0.24 SMART proved its stability and reliable high-accuracy at the global scale.

It is worth mentioning that the visual-check-based accuracy assessment mainly considered the commission error, while omission error cannot be effectively revealed for the following reason. The omitted fires were mainly caused by the requirement of a minimum burning area. Since the spatial resolution of FY-3D and MODIS active fire products is 1 km, small fires (less than $100 \, \mathrm{m}^2$) could not be captured by sensors and recognized through visual check. Meanwhile, thermal abnormalities were seen at the edge of cloud and water bodies, which could be recognized through visual check. In this case, the visual-check-based accuracy assessment mainly considered the commission errors.

## 4.2 Cross-verification between FY-3D and MODIS global fire products

The cross-verification between FY-3D fire products and the mainstream MODIS fire products, MYD14A1 V6 (https://firms.modaps.eosdis.nasa.gov/map/TS14) with a daily temporal resolution and 1 km spatial resolution, was conducted using all the 2019 datasets. The datasets with observation times of less than 1 h were selected; the underlying surfaces were visually checked to remove areas covered by non-vegetation such as water, ice and snow, and bare land. According to the criterion that the distance matching between the two fire spot pixels was less than 0.03°, cross-verification was conducted with different months, underlying surfaces, regions and fire intensities. In 2019, there were 2 237 714 fire spot pixels in

MODIS fire products, 1 866 920 of which were matched with FY-3D fire products, with an overall consistence of 84.4 % (as shown in Fig. 6). As shown in Fig. 6, global fire spots were mainly distributed in America, south-central Africa, East Asia and Southeast Asia, Australia, and parts of Europe, and there were notable spatiotemporal variations in identified fire spots. Specifically, given the overall data volume and spatial distribution, the total number of fire spot pixels from MODIS fire products was larger than that of FY-3D products. For individual regions, the more fire spots, the higher consistence between FY and MODIS fire products. Africa is the region with the most fire spots across the globe. From May to October, a majority of fire spots was located in southern Africa, while a majority of fire spots from November to the next April was located in the middle and western coastal regions of Africa. The consistence between MODIS and FY-3D products was higher than in other regions. The distribution of fire spots in South America also presented seasonal characteristics. From July to October, fire spots were mainly concentrated in the middle parts of South America. For other seasons, fire spots in South America were mainly concentrated in the north and other parts. The consistence between MODIS and FY-3D fire products also demonstrated seasonal differences, with a high consistence from August to November and a relatively low consistence in other seasons. For Eurasia, there were notable seasonal variations in spatial patterns of fire spots. During March to August, there were relatively many fire spots and the consistence between MODIS and FY-3D fire products was relatively high in this region.

In addition to the overall consistence between MODIS and FY-3D fire products, we also conducted cross-verification between the two global fire products in different months, underlying surfaces, regions and fire intensities as follows.

### 4.2.1 Cross-verification between MODIS and FY-3D in different months

Figure 7a illustrates the monthly consistence between FY-3D and MODIS fire products in 2019. The consistence in the remaining months is over 80 % except in April, October and November. The highest appears in July, exceeding 90 %, while the lowest is in April, at 71 %. Detailed parameters can be found in Table 5. From a global perspective, the number of fire spots was larger in July, August and September and the mean consistence between MODIS and FY-3D fire products was larger than 85 %. For July when the fire products were the most numerous, the consistence achieved 90 %. From January to May, the number of fire spots was relatively small, and the mean consistence was around 80 %. The consistence for April was 71 %, the lowest among all months. The notable monthly variations in the consistence between MODIS and FY-3D fire products was mainly attributed to the uneven spatial distribution of fire spots across the globe. As shown in Fig. 6, in June and July, a large number of fire spots were mainly concentrated in Africa, South

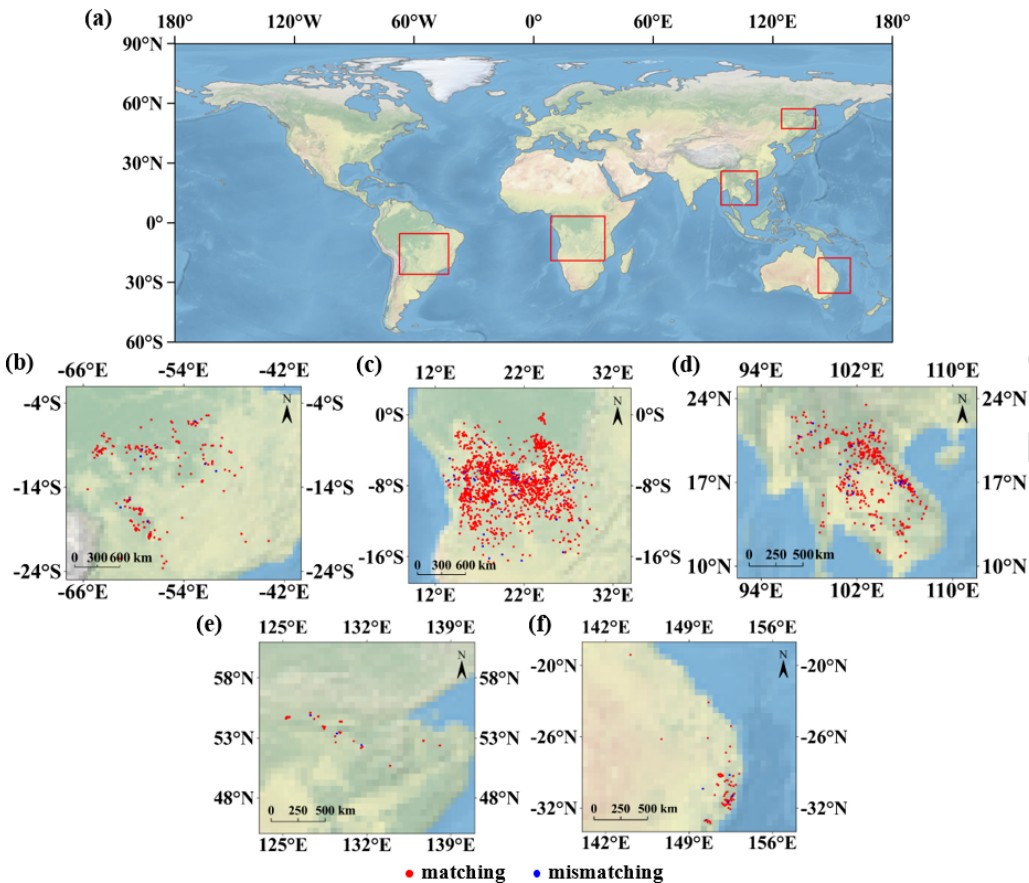

**Figure 5. (a)** Observation positions from FY-3D MERSI-II. The red frame at the upper right shows FY-3D MERSI-II is located at the border between northeast China and Russia. The lower left red frame shows FY-3D MERSI-II is over east-central South America, and the central red frame shows FY-3D MERSI-II is located in south-central Africa. The middle right red frame shows the FY-3D MERSI-II is over the Indochinese Peninsula, and the lower right red frame shows the FY-3D MERSI-II is located in east Australia. **(b–f)** Fire spot matching diagram between GFR and visual interpretation data of FY-3D MERSI-II. The red points indicate that GFR matches visual interpretation data, and the blue points represent that only GFR recognized the fire spots, which was not CE12.

**Table 4.** Accuracy assessment of FY-3D-identified fires based on SMART (visual check).

| Region | GFR-based fire spots | Not matched with SMART | Accuracy (%) |
|---|---|---|---|
| South-central Africa | 1429 | 77 | 94.6 |
| East-central South America | 204 | 12 | 94.1 |
| Siberia | 32 | 3 | 90.6 |
| Australia | 85 | 7 | 91.8 |
| Indochinese Peninsula | 438 | 32 | 92.7 |
| Overall | 2188 | 131 | 94.0 |

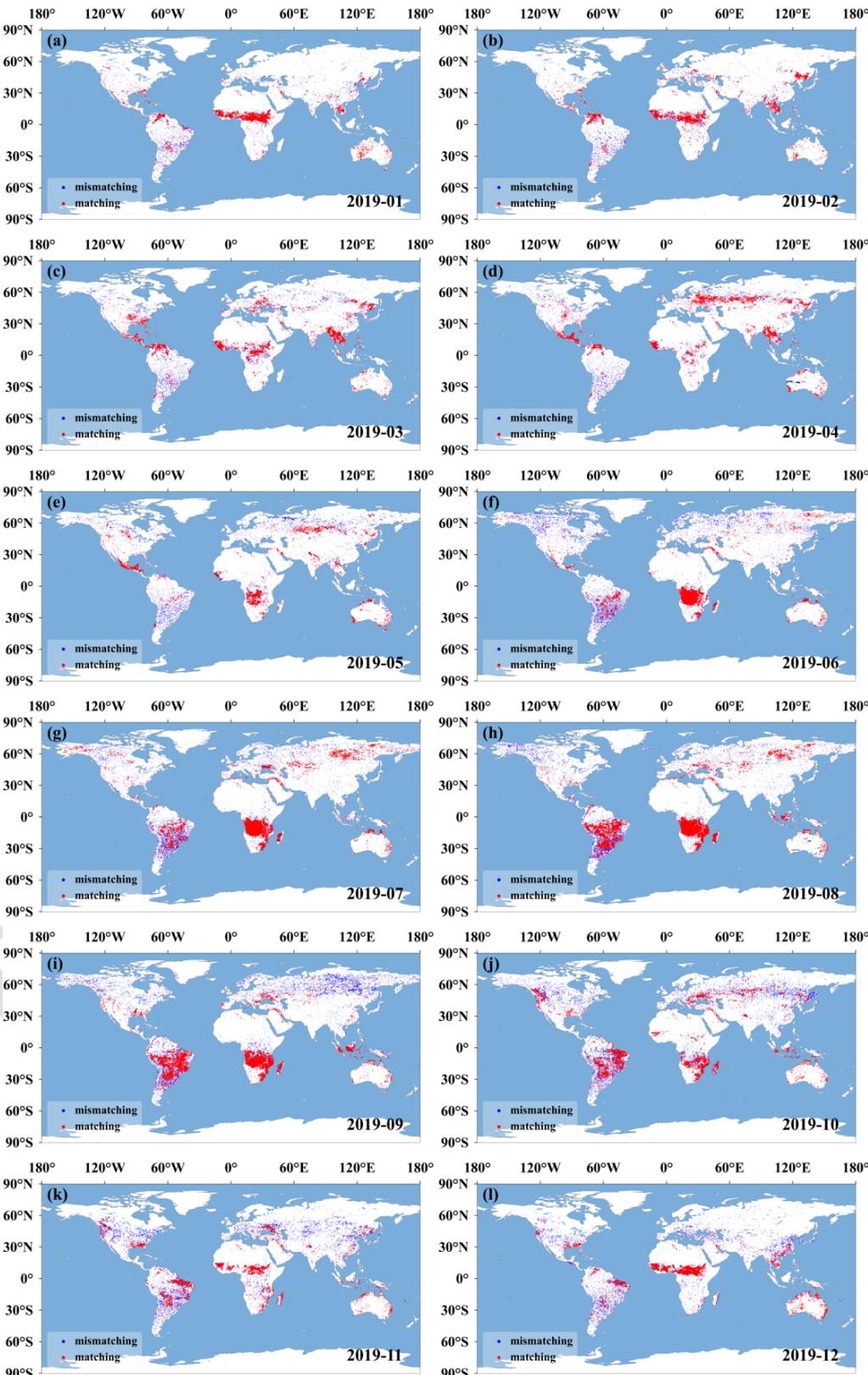

**Figure 6.** The consistence between FY-3D and MODIS fire products in different months (2019).

**Table 5.** Cross-satellite comparison between FY-3D and MODIS fire products.

| Time | Match | Mismatch | Total | Consistence (%) |
|------|-------|----------|-------|-----------------|
| 201 901 | 70 799 | 14 188 | 84 987 | 83 |
| 201 902 | 66 849 | 14 717 | 81 566 | 82 |
| 201 903 | 105 176 | 22 576 | 127 752 | 82 |
| 201 904 | 94 474 | 39 250 | 133 724 | 71 |
| 201 905 | 75 703 | 17 135 | 92 838 | 82 |
| 201 906 | 174 587 | 33 862 | 208 449 | 84 |
| 201 907 | 362 108 | 39 683 | 401 791 | 90 |
| 201 908 | 315 182 | 51 627 | 366 809 | 86 |
| 201 909 | 226 363 | 47 607 | 273 970 | 83 |
| 201 910 | 115 975 | 33 956 | 149 931 | 77 |
| 201 911 | 102 240 | 27 732 | 129 972 | 79 |
| 201 912 | 157 464 | 28 461 | 185 925 | 85 |
| Total | 1 866 920 | 370 794 | 2 237 714 | 83.4 |

America and Eurasia, leading to a high consistence of fire identification. In April, there were limited and sparsely distributed fire spots in Africa and South America, leading to a low consistence. According to the statistics, the number of fire spots was positively correlated with the consistence between different fire products. Meanwhile, in seasons when fire could last longer, the consistence was higher.

### 4.2.2 Cross-verification between MODIS and FY-3D on different underlying surfaces

Statistical analysis of consistence is carried out with different types of underlying surface. The data of underlying surfaces are according to the global land use detailed in Table 6.

The 15 types of underlying surfaces were selected for verification. Table 6 and Fig. 7c show the consistence of FY-3D and MODIS fire products with different underlying surfaces. From the classification of different underlying surfaces, the remaining types are over 80 % consistent except (11) Post-flooding or irrigated croplands (or aquatic), (14) Rainfed crops, (20) Mosaic cropland (50 %–70 %)/vegetation (grassland/shrubland/forest) (20 %–50 %), (140) Closed to open (> 15 %) herbaceous vegetation (grassland, savannas or lichens/mosses), and (150) Sparse (< 15 %) vegetation. When the underlying surface is open (15 %–40 %) coniferous and deciduous forest or evergreen forest, the consistence is the highest, at 93 %. In addition, according to the classification of underlying surfaces, the fire spot identification shows high consistence when the underlying surface is forest. The consistence between FY-3D and MODIS fire spots on different underlying surfaces in each month is demonstrated in Table 7. Clearly, we can find the fluctuation in consistence across seasons due to the variation in combustible vegetation, which influenced the detecting capability of MODIS and FY-3D.

The low consistence between FY-3D and MODIS fire products was observed for underlying surfaces 11, 14, 20, 140 and 150. Specifically, 11, 14 and 20 could be categorized as farmlands. Surface 140 was mainly occupied by herbaceous vegetation or sparse grasslands. Surface 150 was mainly occupied by sparse grasslands. Generally, these surfaces were all covered by sparse or unstable vegetation, on which the fire can last for a relatively short period. Meanwhile, the observation time lag between FY-3D and MODIS was larger than 30 min. Therefore, the consistence of FY-3D and MODIS fire products on these surface types was lower than on other surface types.

### 4.2.3 Cross-verification between MODIS and FY-3D in different regions

The global monitoring area is divided into Africa, America, Asia, Europe and Oceania. The verification demonstrates the results with the highest consistence (over 80 %) are found in Africa and Asia, and those in America, Europe and Oceania show consistence over 70 %. The FY-3D MERSI-II fire identification algorithm draws lessons from the MODIS algorithm and has been improved on that basis, and targeted development has been made for the underlying surface and climatic conditions in China, so it is necessary to test the matching results in China separately. This shows that China's regional consistency CE13 of results is lower than for other continents, at only 65 %. To further examine the suitability of FY-3D fire products in China, an accuracy assessment of FY-3D and MODIS fire products was conducted based on ground truth data and is explained in the following sections.

### 4.2.4 Cross-verification of MODIS and FY-3D in terms of fire intensities

The confidence of fire spots and the fire intensity represented by FRP are analyzed, and the data come from the MODIS fire spot list. Figure 8a and b are statistical diagrams of confidence and FRP, respectively. From Fig. 8a, the confidence of the matched pixels of the two satellites is above 66 %, while that of the mismatched ones is less than 60 % and even lower than 50 % in some months. In other words, the higher confidence indicates the higher matching degree. As indicated by Fig. 8b, the FRP of the matched pixels of two satellites is mostly above 40 MW, while that of the unmatched pixels is less than 40 MW and even lower than 20 MW in some months. Accordingly, the greater fire intensity leads to a greater probability of simultaneous observation by the two satellites and a higher matching degree between their results.

Two major findings were identified based on the comparison between FY-3D and MODIS fire products in terms of fire intensity: firstly, the higher the credential of the identified fire, the higher consistence between FY-3D and MODIS fire products. When the credential was larger than 65 %, both FY-3D and MODIS could effectively identify the candidate

**Table 6.** Classification of underlying surfaces (land cover types).

| ID | Definition of underlying surfaces |
| --- | --- |
| 11 | Post-flooding or irrigated croplands (or aquatic) |
| 14 | Rainfed croplands |
| 20 | Mosaic cropland (50 %–70 %)/vegetation (grassland/shrubland/forest) (20 %–50 %) |
| 30 | Mosaic vegetation (grassland/shrubland/forest) (50 %–70 %)/cropland (20 %–50 %) |
| 40 | Closed to open (> 15 %) broadleaved evergreen or semi-deciduous forest (> 5 m) |
| 50 | Closed (> 40 %) broadleaved deciduous forest (> 5 m) |
| 60 | Open (15 %–40 %) broadleaved deciduous forest/woodland (> 5 m) |
| 70 | Closed (> 40 %) needleleaved evergreen forest (> 5 m) |
| 90 | Open (15 %–40 %) needleleaved deciduous or evergreen forest (> 5 m) |
| 100 | Closed to open (> 15 %) mixed broadleaved and needleleaved forest (> 5 m) |
| 110 | Mosaic forest or shrubland (50 %–70 %)/grassland (20 %–50 %) |
| 120 | Mosaic grassland (50 %–70 %)/forest or shrubland (20 %–50 %) |
| 130 | Closed to open (> 15 %) (broadleaved or needleleaved, evergreen or deciduous) shrubland (< 5 m) |
| 140 | Closed to open (> 15 %) herbaceous vegetation (grassland, savannas or lichens/mosses) |
| 150 | Sparse (< 15 %) vegetation |

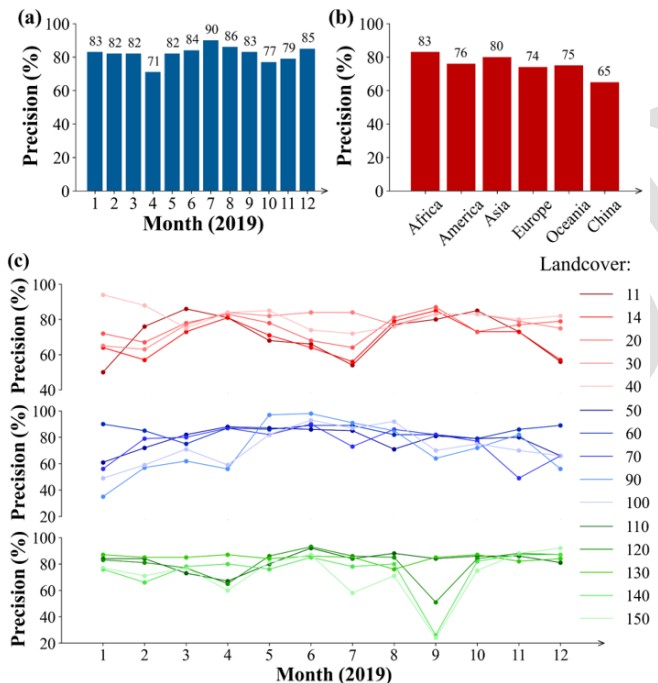

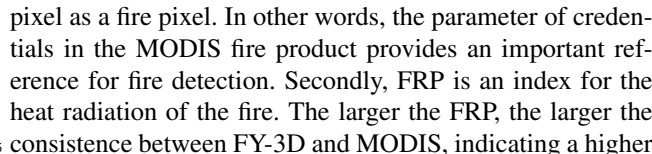

**Figure 7.** Consistence between FY-3D and MODIS fire products under different conditions. **(a)** Consistence between FY-3D and MODIS fire products in different months. **(b)** Consistence between FY-3D and MODIS fire products in different regions. **(c)** Consistence between FY-3D and MODIS fire products on different underlying surfaces.

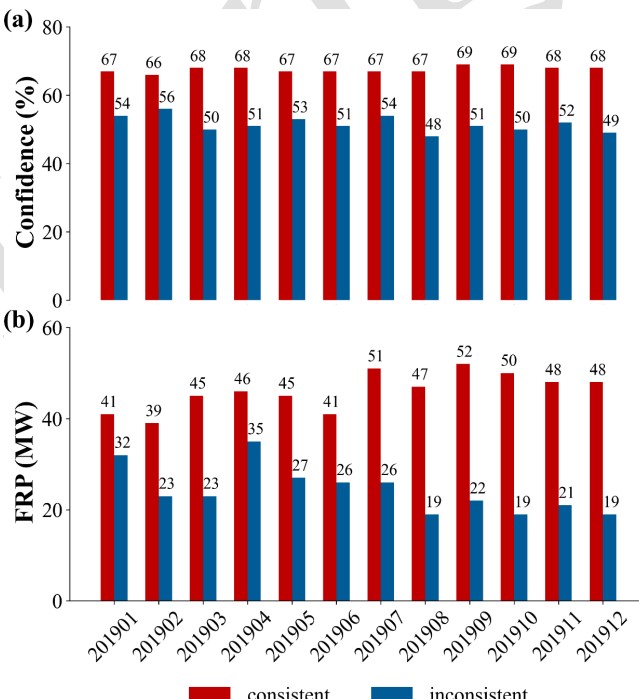

**Figure 8. (a)** Confidence of consistent and inconsistent pixels between FY-3D and MODIS fire products. **(b)** FRP of consistent and inconsistent pixels between FY-3D and MODIS fire products.

pixel as a fire pixel. In other words, the parameter of credentials in the MODIS fire product provides an important reference for fire detection. Secondly, FRP is an index for the heat radiation of the fire. The larger the FRP, the larger the consistence between FY-3D and MODIS, indicating a higher accuracy of fire detection. Therefore, the difficulty for fire detection mainly lies in the detection of weak fires.

## 4.3 Accuracy assessment of FY-3D fire products in China based on field-collected reference

In addition to visual check and consistence check, we also referred to a large-scale field experiment to comprehensively

**Table 7.** The consistence between FY-3D and MODIS fire spots on different underlying surfaces in each month (total FY-3D pixels(consistence).

| ID | Jan | Feb | Mar | Apr | May | Jun | Jul | Aug | Sep | Oct | Nov | Dec |
|---|---|---|---|---|---|---|---|---|---|---|---|---|
| 11 | 754(50 %) | 1471(76 %) | 1651(86 %) | 450(81 %) | 201(68 %) | 344(66 %) | 353(54 %) | 678(77 %) | 1786(80 %) | 1516(85 %) | 558(73 %) | 416(56 %) |
| 14 | 4459(64 %) | 5024(57 %) | 7745(73 %) | 11 439(81 %) | 6818(71 %) | 4137(64 %) | 2135(56 %) | 4122(79 %) | 8090(85 %) | 4561(73 %) | 3154(73 %) | 1663(57 %) |
| 20 | 8033(72 %) | 8596(67 %) | 13 513(78 %) | 20 282(83 %) | 14 772(78 %) | 5216(68 %) | 2921(64 %) | 5449(81 %) | 11 970(87 %) | 5858(73 %) | 4721(77 %) | 5572(79 %) |
| 30 | 5786(65 %) | 7227(63 %) | 13 018(77 %) | 22 626(84 %) | 26 523(82 %) | 23 024(84 %) | 16 007(84 %) | 6455(77 %) | 14 534(83 %) | 16 523(83 %) | 8646(79 %) | 5199(75 %) |
| 40 | 45 313(94 %) | 38 194(88 %) | 25 315(75 %) | 63 474(84 %) | 69 987(85 %) | 14 770(74 %) | 8265(72 %) | 7107(76 %) | 22 921(83 %) | 31 839(83 %) | 14 646(80 %) | 9556(82 %) |
| 50 | 3454(61 %) | 8398(72 %) | 19 960(82 %) | 45 387(88 %) | 51 148(87 %) | 42 981(86 %) | 25 424(85 %) | 4356(71 %) | 5481(81 %) | 6237(79 %) | 3713(80 %) | 1920(66 %) |
| 60 | 36 987(90 %) | 6321(85 %) | 5570(75 %) | 25 021(87 %) | 49 083(86 %) | 74 660(89 %) | 59 345(89 %) | 3028(86 %) | 6526(82 %) | 4478(79 %) | 12 513(86 %) | 18 192(89 %) |
| 70 | 1863(56 %) | 3655(79 %) | 5031(80 %) | 4052(87 %) | 1865(82 %) | 3411(90 %) | 2123(73 %) | 3402(85 %) | 2346(82 %) | 2791(77 %) | 704(49 %) | 719(66 %) |
| 90 | 840(35 %) | 3255(57 %) | 8901(62 %) | 11 125(56 %) | 61 299(97 %) | 135 344(98 %) | 32 767(91 %) | 18 539(85 %) | 4645(64 %) | 4076(72 %) | 1484(82 %) | 608(56 %) |
| 100 | 1079(49 %) | 1851(59 %) | 3423(71 %) | 1988(59 %) | 2444(82 %) | 6027(93 %) | 3677(87 %) | 8695(92 %) | 2813(70 %) | 2596(75 %) | 565(70 %) | 397(66 %) |
| 110 | 19 896(84 %) | 13 825(84 %) | 4194(73 %) | 3669(67 %) | 6504(80 %) | 11 351(92 %) | 7407(84 %) | 7223(88 %) | 4268(84 %) | 4983(86 %) | 5009(86 %) | 5409(81 %) |
| 120 | 6568(83 %) | 3406(81 %) | 3639(77 %) | 3602(65 %) | 9037(86 %) | 12 972(93 %) | 7122(86 %) | 4999(85 %) | 3574(51 %) | 2379(84 %) | 4651(88 %) | 4710(87 %) |
| 130 | 38 258(87 %) | 18 784(85 %) | 19 935(85 %) | 34 627(87 %) | 37 668(84 %) | 34 189(86 %) | 20 881(85 %) | 6963(76 %) | 20 071(85 %) | 27 134(87 %) | 8320(82 %) | 15 465(84 %) |
| 140 | 3941(76 %) | 2905(66 %) | 6159(78 %) | 7692(80 %) | 6756(76 %) | 8964(85 %) | 5139(78 %) | 3104(80 %) | 13 060(26 %) | 3562(82 %) | 3844(87 %) | 4270(87 %) |
| 150 | 5760(77 %) | 5073(71 %) | 8872(77 %) | 7268(60 %) | 15 938(81 %) | 19 370(87 %) | 10 467(58 %) | 4106(71 %) | 12 991(24 %) | 3532(75 %) | 6359(88 %) | 8994(92 %) |

assess the suitability of FY-3D fire products in China. The State Grid Corporation of China and China Meteorological Administration jointly conducted a fire detection experiment throughout 2020 in five provinces in China: Guangdong, Guangxi, Yunnan, Guizhou and Hainan. This experiment was conducted in the following steps. A large number of drones were employed to check the occurrence of fires. According to the local passing time of FY-3D, these drones reported the coordinates of actual fires for verifying the accuracy of FY-3D-identified fires. The temporal difference between the passing time of FY-3D and reported time was controlled to within 1 h. In this way, both omitted and misidentified fires could be effectively recognized (as shown in Fig. 9). Based on the field-collected reference of fires, we evaluated the suitability of FY-3D fire products in China (Table 8).

As shown in Fig. 9 and Table 8, FY-3D products achieved a good accuracy of 79.43 % in China. Meanwhile, MODIS also achieved a good accuracy of 74.23 %. As introduced above, the omission error in FY-3D and MODIS fire products was mainly attributed to a small fire area, which failed to meet the minimum fire area recognizable by sensors. When simply considering the commission error, FY-3D fire products achieved an accuracy of 88.50 %, notably higher than that of MODIS (79.69 %). This result proved that with the consideration of local underlying surfaces, FY-3D fire products are more suitable for fire monitoring in China.

## 5 Discussion

### 5.1 Advantages, limitations and implementations of FY-3D fire products

As satellite instruments keep aging in the harsh space environment, the degradation of sensors is inevitable (Tian et al., 2015). Theoretically, sensor degradation can be corrected through atmospheric calibration. However, during the mission life, the solar diffuser and stability monitor required for atmospheric calibration also change across time (Wang et al., 2012). Since the MODIS instrument has been working for more than 20 years, its performance for fire detection will degrade, if it has not already, in the future. Furthermore, similarly to VIIRS and other algorithms, MODIS fire products may have large uncertainties in such regions as China (Fu et al., 2020; [TS15] Ying et al., 2019 [TS16]).

As major products of the FY-3D meteorological satellite, FY-3D fire products boast a high resolution and accuracy in China by specifically including the underlying surface parameters collected in China. Compared with MODIS and VIIRS, MERSI-II shows a resolution of 250 m in the far-infrared channel, which is the highest among meteorological satellites of the same type. The FY-3D fire identification algorithm learns from the advantages and technical ideas of MODIS and VIIRS fire identification algorithms. Furthermore, FY-3D fire products have been optimized in terms of

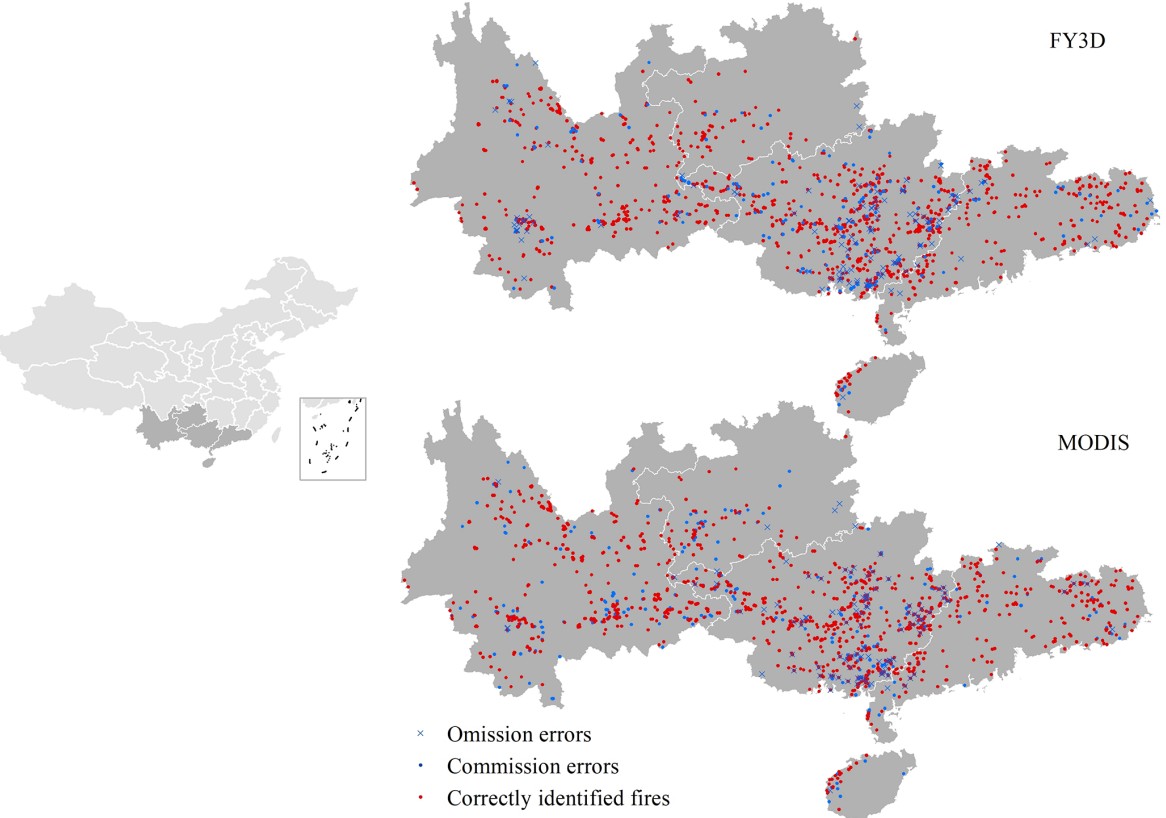

**Figure 9.** Accuracy assessment of FY-3D fire products in China based on the ground-based reference.

**Table 8.** Accuracy assessment based on field ground truth.

|  | Correct identification | Omission | Commission | Accuracy (%) | Accuracy without omission (%) |
|---|---|---|---|---|---|
| FY-3D | 1178 | 133 | 172 | 79.43 % | 88.50 % |
| MODIS | 1201 | 112 | 306 | 74.23 % | 79.69 % |

auxiliary parameters, fire identification and re-identification as follows.

*Auxiliary parameters.* Since the sole use of the vegetation index is limited to reflecting combustible materials, climatic boundaries and geographical environment data, which have a strong influence on vegetation types and growth, are added to FY-3D fire identification.

*Fire identification.* FY-3D adopts the adaptive threshold and reduces the limitations caused by fixed thresholds of MODIS and VIIRS algorithms. Meanwhile, FY employs a re-identification index according to geographical latitude and underlying surface types, as well as the influence by cloud, water bodies and bare land, and the comprehensive consideration of multiple influencing factors increases the accuracy of fire identification. Thirdly, since the far-infrared channel plays an important role in fire identification and FY-3D has a

high resolution of 250 m in that channel, the accuracy of fire identification is improved.

*Fire re-identification.* FY-3D fire products can be used for both global climate change research and such practical implementations as forest and grassland fire prevention with a higher requirement for accuracy. Based on the initially identified fire spots, FY-3D employed the re-identification index to further remove false fire spots at cloud edges, water body edges and other high-reflection underlying surfaces.

The MODIS fire product is one of the most significant and frequently employed fire products with mature algorithms. Compared with MODIS, FY-3D receives limited emphasis for its capability for fire monitoring, which is mainly attributed to its short service periods. On one hand, due to its long time series and general reliability, MODIS fire products have remained a popular choice for monitoring long-term variations in fire spots across the world. However, the

long-term running of MODIS sensors has led to growing uncertainties about the quality of recent and future MODIS fire products. In this regard, thanks to its similar spatiotemporal resolution, high consistence and visiting time difference of less than 1 h, FY-3D fire products have the potential to be widely employed as a potential alternative to and continuity for global MODIS fire products. Meanwhile, FY-3D fire products have a higher reliability in China and its surrounding regions than other fire products. Therefore, FY-3D fire products are an ideal selection for fire monitoring in China.

The main implementation of FY-3D fire products is fire monitoring. For vast forest and grassland areas, it is inefficient and time-consuming for manual and aircraft patrols to monitor wildfires. Satellite remote sensing can work for a continuous space with a wide monitoring range, providing massive amounts of information in fire detection, disaster relief and post-disaster assessment. In addition to fire spot identification and real-time fire tracking, the impact of pollutants produced by biomass combustion on the environment is another important topic. In China and Southeast Asia, air pollution caused by biomass burning has intensified in recent years. Agricultural activities such as crop residue burning and wildfires (e.g., forest fires and grassland fires) emit airborne pollutants (e.g., $PM_{2.5}$, $PM_{10}$, CO). In this regard, FY-3D fire products can be used as the emission sources for estimating their environmental effects.

## 5.2 Future extension of FY-3D fire products

China recently launched the FY-3E and FY-4B satellites in June and July 2021. Amid the launch and operation of a new generation of Fengyun meteorological satellites, the accuracy and timeliness of fire monitoring by meteorological satellites have been largely enhanced. Thanks to improved meteorological data, which provide a useful reference to understand the current status of combustibles and potential fire risk, the FY-3D satellite will be taken as a better data source to produce various secondary products for fire monitoring and prediction. Based on traditional fire spot identification, further research should concentrate on the assessment of the fire area, estimation of biomass carbon emissions, prediction of smoke impact, and early warning of forest and grassland fire using the series of Fengyun meteorological satellites. For instance, the water content of combustibles is closely related to temperature, light and cloud cover, which are important indicators in forest and grassland fire forecasts. However, this variable has rarely been considered in previous fire products. Based on a series of products from Fengyun meteorological satellites, such as the surface temperature, vegetation index, surface evapotranspiration, solar radiance and cloud cover, FY-3D fire products can be improved by establishing an estimation model for the water content of combustibles. Meanwhile, with fire products such as fire spots and smoke and meteorological products such as wind field data from Fengyun series satellites, we can predict the impact of smoke caused by forest and grassland fires on the atmospheric environment in the surrounding areas. In future implementations, Fengyun meteorological satellites will play a greater role in monitoring, forecasting and early warning of global fires and their ecological impacts.

## 6 Data availability

MYD14A1 Version 6 is available via the NASA FIRMS portal (https://firms.modaps.eosdis.nasa.gov/map/, NASA FIRMS, 2021). FY-3D fire products are now downloadable from our official website (http://satellite.nsmc.org.cn/portalsite/default.aspx, NSMC, 2021) using a registered account and password. For the convenience of data checking and trial experiments, a test account is provided with the account name "1256931756@qq.com" and password "yangjing1211".

## 7 Conclusions

With a similar spatial and temporal resolution, we produced FY-3D global fire products, aiming to serve as a potential alternative to and continuity for MODIS fire products. The sensor parameters and major algorithms for noise detection and fire identification in FY-3D products were introduced. For visual-check-based accuracy assessment, five typical regions across the globe, Africa, South America, the Indochinese Peninsula, Siberia and Australia, were selected, and the overall accuracy exceeded 94 %. We also compared the FY-3D and MODIS fire products for their consistence. The result suggested that the overall consistence was 84.4 %, with fluctuation across seasons, surface types and regions. The high accuracy and consistence with MODIS products proved that the FY-3D fire product is an ideal tool for global fire monitoring. Based on field-collected reference data, we further evaluated the suitability of FY-3D fire products in China. The overall accuracy and accuracy without considering omission errors were 79.43 % and 88.50 % higher, respectively, than those of MODIS fire products. Since detailed geographical conditions in China were considered, FY-3D products should be preferably employed for monitoring fires and estimating their environmental effects in China.

**Author contributions.** JC, WZ and CL produced FY-3D global fire products and the official website. JC, ZC, BG and ML conceived the manuscript. JC, CZ, QY, MX, XC and JY conducted data analysis and produced figures. JC and ZC wrote the draft. ZC and ML reviewed and revised the manuscript.

**Competing interests.** The contact author has declared that neither they nor their co-authors have any competing interests.

**Acknowledgements.** . TS17

**Financial support.** This research has been supported by the National Natural Science Foundation of China (grant no. 42171399) and the National Key Research and Development Program of China (grant no. 2021YFC3000300). TS18

**Review statement.** This paper was edited by Bo Zheng and reviewed by two anonymous referees.

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

## Remarks from the language copy-editor

CE1     Please note the edits to this affiliation.

CE2     Please note this manuscript has undergone copy-editing according to the standards of American English and serial (Oxford) commas have not been applied, except in the case of complex lists (i.e., those with items including "and" or "or").

CE3     Please check the use of "typical" in this context throughout the paper. Do you mean something like "commonly delineated regions"?

CE4     This abbreviation is not defined. Is it well known, or should it be defined for clarity?

CE5     Please confirm this is the official name of the system. If it is a generic term, it should be lowercase instead.

CE6     Please note edits to this table carefully as they may not show up in the track-changes PDF.

CE7     Please check the following phrase. The meaning is unclear.

CE8     Please check the following text. It is a fragment, and the meaning is unclear.

CE9     Please check whether "bright temperature(s)" should be "brightness temperature(s)" throughout.

CE10     What does the lowercase "k" mean in the two instances here? We do not allow "k" to stand for 1000 on its own.

CE11     Please confirm this is an official name of a specific system. If it is a generic term, it should be lowercase.

CE12     Please check. Either a word or phrase is missing here, or "which was not" should be deleted.

CE13     Elsewhere "consistence" is used. The "consistency" variant is more common, but both are fine. Would you like them to be made the same throughout (i.e., this instance changed to "consistence" or all other instances changed to "consistency")?

## Remarks from the typesetter

TS1     The composition of Figs. 4, 7 and 8 has been adjusted to our standards.

TS2     Please add reference to reference list.

TS3     Please provide date of last access.

TS4     Please add reference to reference list.

TS5     Please add reference to reference list.

TS6     Please add reference to reference list.

TS7     Please provide date of last access.

TS8     Please check table; it is unclear which rows in the Application column belong together.

TS9     Please confirm unit.

TS10     Is the dot a multiplication sign or should it denote a decimal point (1.0)?

TS11     Please check.

TS12     Please add reference to reference list.

TS13     Please confirm change.

TS14     Please provide date of last access.

TS15     Please add reference to reference list.

TS16     Please add reference to reference list.

TS17     Would you like to add acknowledgements?

TS18     Please note that the funding information has been added to this paper. Please check if it is correct. Please also double-check your acknowledgements to see whether repeated information can be removed or changed accordingly. Thanks.

TS19     Please ensure that any data sets and software codes used in this work are properly cited in the text and included in this reference list. Thereby, please keep our reference style in mind, including creators, titles, publisher/repository, persistent identifier, and publication year. Regarding the publisher/repository, please add "[data set]" or "[code]" to the entry (e.g., Zenodo [code]).

TS20     Please provide article number or page range.

TS21     Reference not cited in the text.

TS22     Reference not cited in the text.

TS23     Reference not cited in the text.

TS24     Reference not cited in the text.

TS25     Please provide article number or page range.

TS26     Please provide a persistent identifier.

TS27     Please provide more information.

TS28     Reference not cited in the text.