# Peer review of "The FY-3D Global Active Fire product: Principle,"

_Earth System Science Data, 2022_

## Author Comment (AC1)

Thanks very much for your interest.

Hi, Line 361,Formula (5):

1.What is the unit of $(P*S\lambda,\varphi)$ ? In the dataset of FY-3D active fire product( Provided by NSMC), there is a field of AreaFire. Does "AreaFire" equal to $(P*S\lambda,)$ ?

**R: In formula (5), $(P*S\lambda,\varphi)$ stands for the area of burning fire, so its unit is m². And yes, Areafire equal to $(P*S\lambda,\varphi)$.**

2.The FRP value of each point was not provided by the raw dataset directly. Could you show the FRP value of some pixels or provide the table of fire point FRP used to generate Fig8b with the FY-3D active fire product(2019)?

3.How about the spatial distribution of fire FRP?

**R: Thanks so much for your interest in FRP. Actually, the sole demonstration of pixel-level FRP provided limited information. Instead, for Fig 8, by comparing FRP between MODIS and FY-3D products, two major findings were identified based on the comparison between FY-3D and MODIS fire products in terms of fire intensity: Firstly, the higher the credential of the identified fire, the higher consistence between FY-3D**

and MODIS fire products. When the credential was larger than 65%, both FY-3D and MODIS could effectively identify the candidate pixel as fire pixel. In other words, the parameter of credential in MODIS fire product provides important reference for fire detection. Secondly, FRP is an index for the heat radiation of the fire. The larger FRP, the larger consistence between FY-3D and MODIS was, indicating a higher accuracy of fire detection.

So in the manuscript, we did not demonstrate the distribution of FRP. Here, we could present the monthly distribution of global FY-3D FRP in 2019 for your reference. Thanks again for your interest.

January

February

0≤FRP<25
25≤FRP<50
50≤FRP<75
75≤FRP<100
FRP≥100

March

April

[Figure]

[Figure]

September

October

[Figure]

November

December

---

## Author Comment (AC2)

RC1:It is well known that the quality of commonly used AQUA MODIS global fire products inevitably decrease gradually after a long-term running. It is very glad to see that Chen et al. made a robust global fire products with similar spatiotemporal resolution to AQUA MODIS products, which guarantees the continuity of research based on MODIS data. Overall, this manuscript is well-structured and presents sufficient details. The verification of FY-3D fire product was conducted from different perspectives. The figures were presented with good quality. Nevertheless, the following several technique issues can be considered by the authors before I recommend its final acceptance for publication in ESSD:

**To Referee 1:**

**Thanks so much for your encouragement and valuable comments. We have revised the manuscript according to your general and detailed comment. Please feel free to contact us if additional revisions are required.**

1 What is the major difference between the algorithm of fire identification for FY-3D and MODIS fire products? Although briefly introduced, more detailed explanation should be added. This can give readers and users a better picture of the quality and advantage of FY-3D fire products.

**R: This is a good point and thanks for pointing it out. The principle for identifying fires based on FY-3D and MODIS is generally the same. As explained in the manuscript, FY-3D fire products fully considered the advantage of MODIS fire products and made some further improvement as follows:**

**(1) Auxiliary parameters:Since the sole use of vegetation index is limited to reflect combustible materials, climatic boundaries and geographical environment data, which had a strong influence on vegetation types and growth, were added to FY-3D fire identification.**

**Fire identification: FY-3D adopts the adaptive threshold and reduces the limitations caused by fixed thresholds of MODIS and VIIRS algorithms. Meanwhile, FY employs a re-identification index according to geographical latitude, underlying surface types, as well as the influence by cloud, water bodies and bare land and the comprehensive consideration of multiple influencing factors increases the accuracy of fire identification; Thirdly, since the far-infrared channel plays an important role in fire identification and FY-3D has a high resolution of 250 m in the far-infrared channel, The precision of fire identification is improved.**

**Fire re-identification: FY-3D fire products can be used for both global climate change research and such practical implementations as forest and grassland fire prevention with a higher requirement for precision. Based on the initially identified fire spots,**

**FY-3D employed the re-identification index to further remove fire spots at cloud edges, cloud gaps, water body edges, and conventional heat sources and on bare land and highly reflective underlying surfaces.**

**This part was concluded in the discussion part.**

2 There is a difference of observation time between FY-3D and MODIS fire products. Will this cause potential errors when employing both FY-3D and MODIS products? e.g. the first part of time series based on MODIS product and the latter part based on FY-3D products?

**R: Thanks for your comment. This is a good point. Yes, for the same period, different observation time may lead to some difference for the identified fires and the influence of different observation time on the difference of observed fires varied across different FRP. As introduced in the manuscript, the larger FRP (or denser vegetation cover), the longer fire existed, the better consistence between two fire products. Currently, the temporal difference of observation time between MODIS and FY-3D was the smallest and notably smaller than that between MODIS and other fire products. Therefore, the relative small difference between MODIS and FY-3D exerted a very limited influence on the consistence of two fire products, and FY-3D fire products were so far the best continuity for current MODIS products.**

3 There are two middle-infrared band (3.8um and 4.05um). What is the difference between the two bands? And what is major advantage (e.g. detecting specific fires?) for the two bands? What is the band used for FY-3D fire products?

**R: This is a good comments. Actually, both middle-infrared band (3.8um and 4.05um) were sensitive to strong heat signals. Their difference lied in their performance under different temperature and radiation conditions. 3.8um was more close to the wavelength of solar radiation, and had better reflection under solar radiation. As a comparison, 4.05um was more easily to miss weak fires. Therefore, current FY-3D fire products were produced based on 3.8um band for better fire identification. In the future, we would consider the comprehensive use of both bands for improved fire products. Thanks so much for your comment. Relevant description has been added to the revised manuscript.**

4 The position accuracy for FY-3D and MODIS fire products were both 0.01°. Why the range for matching them was set as 0.02°?

**R: This is a good point. After projection and resampling, the theoretically optimal resolution was 0.01°. However, there were some bias for satellite positioning and most fires were sub-pixel information, the error tolerance is required. If the resolution was set as 0.01°, the positioning error would cause the same fire pixel recognized as two different fire pixels in FY-3D and MODIS fire products. In this case, we considered a buffer for a pixel and set the range of matching as 0.02°.**

5. On page 16, the equation for position matching suggested 0.03 °, why in the text was 0.02? Is it a typo or something to explain?

**R: We are sorry for this typo. This should be 0.02 °. Thanks for pointing it out.**

6. Page 1 Line 36: Wild fires can significantly affected the formation of cloud and precipitation, which can be mentioned as well. The authors can refer to the following literatures: (a) doi: 10.1029/2021GL094224; (b) doi: 10.1029/2019JD032136 ¼

**R: Thanks for pointing this out. These references have been added.**

7. Page 1 Line 36: Some important references can be added to better support "remarkably deteriorated air quality", such as http://dx.doi.org/10.1080/01431161.2010.485213

**R: Thanks for pointing this out. This reference has been added.**

---

## Author Comment (AC3)

RC2:General comments:

This paper introduced a new data set of active fire using FY-3D satellite. The authors introduced the product specifications, underlying algorithm and notably they compared their product with MYD14A1 MODIS active fire product. I believe such a product, other than MODIS, should surely be welcomed for the community. But unfortunately, I cannot offer its publication in ESSD. However, my views might be biased because of vegetation modeling perspective. Below are my arguments:

**R: Thanks so much for your valuable comments. Clearly, you are a great expert in remote sensing and vegetation modelling and by revising according to your comments, this manuscript has been largely improved. Actually, we would like to emphasize that we are also big fans and have been beneficial significantly from the long time series MODIS data. We are not saying that at the global scale, FY-3D presented a notably better performance than MODIS data, which cannot be proved without reliable global reference data set. Instead, we acknowledged that MODIS fire products presented a generally good accuracy and a more than 20 years' time series, which is no doubt the best source for a diversity of long-term research. However, although it has been the best fire products for twenty years, the gradual ageing of sensors will, if not already, cause the future degradation of MODIS fire products. In this case, if scholars would like to continuously make full use of the long-term series of MODIS fire product, when it degrades significantly or even stops services in the future, a fire product with good reliability and similar characteristics is urgently needed. Against this background, we proposed our FY-3D global fire products as a promising alternatives and continuity for the existing MODIS fire products. To prove the feasibility, we introduced that the spatiotemporal resolution was similar to MODIS fire products, and their observation lag was the smallest compared with other fire products. Meanwhile, we proved its high accuracy and good consistence with MODIS products at the global scale. At the regional scale, we suggested that FY-3D products can have an even better performance than MODIS fire products in China (We are sorry that we did not provide sufficient quantitative evidence in the previous manuscript. And according to your valuable comments, we added some rare ground observation data to the revised manuscript). So in general, we are not suggesting users to fully replace MODIS data with FY-3D fire products. Actually, the long-term historical observation data provided irreplaceable value for the community. We are not questioning the quality of existing MODIS fire products. Instead, we mainly focused on the possible risk of its degradation or stop-service after long-term running. At the global scale, due to their high consistence, FY-3D fire products could serve as reliable post-continuity of MODIS fire products, which could easily integrated with previous long time-series MODIS fire products before 2018. At the regional scale, since better local underlying surface conditions are considered, FY-3D products may be preferably employed in such regions as China.**

**On the other hand, we should say that despite some shared characteristics, the principle and verification for MODIS-based burned-areas and fire products**

presented major differences. A unique challenge for active fire products is how to evaluate the accuracy of the real-time monitoring of occurring fires (instead of the burned areas). For vegetation modelling products or burning areas products, it is much easier to conduct field survey to gain ground truth data for verification, as the observed vegetation or burned areas would remain unchanged for days or weeks. As a comparison, active fire products refers to the fires identified during burning process. In other words, when the fire stopped, the ground observation truth cannot be acquired, as the verification for active fire products lies in the capability of capturing occurring fires through thermal information extracted in corresponding bands. Since active fire lasts in a very short period, it is not feasible to actually stay there and wait for the occurrence of fire at a specific spots. And the active wild burning experiments are highly restricted. In this case, not only for this research, ground observation reference data for active fire products has rarely been employed for all other relevant studies. To verify the reliability of active fire products, there are three possible ways. First and the most frequent employed approach, visual interpretation based on expert knowledge. Second, treat one fire product as reference and compared this reference product with the target product, which is also frequently employed. Third, verification based on ground truth, which is most accurate, yet difficult and rarely employed by previous studies. In the previous manuscript, we employed visual check for quantifying the quality of FY-3D fire products globally. Meanwhile, we consider MODIS fire products as the reference data with good accuracy, and conducted the consistence analysis. In the revised manuscript, despite the extreme difficulty, we applied for some very rare ground observation reference data from a large field firing experiment conducted by China Meteorology Bureau, which further proved the suitability of FY-3D fire products in China. So in general, the active fire products were different from vegetation or burned area products and their verification approaches were limited by the extreme difficulty for acquiring real-time acquired ground truth of observed active fires. However, we have managed to employ all possible approaches (and the large-scale ground truth data were rarely employed) to prove the reliability of FY-3D fire products and its potential to be used as alternatives and future continuities of MODIS data. Thanks so much for raising these issues, which encouraged us to fully revise and significantly improve this manuscript.

More detailed item-to-item responses were presented as follows. Please feel free if additional revisions are required.

- In the background, the authors seem mixing different products of fire mapping. The area actually burned by fire, or burned area products, are the dominant products in terms of fire mapping nowadays used by Earth system science community. Notably, these data can allow estimation of fire emissions for multiple substances, such as $CO_2$, black carbon and aerosols, which are crucial for climate prediction. They also allow for study fire impacts on ecosystems, another critical domain of vegetation fires in the field of Earth system science. The second dominant fire product by satellite mapping, is the one presented here but also made available by MODIS team, that is, active fire mapping.

**R: Thanks so much for this professional comment. Yes, we should made it clear that the active fire products were largely different from burned area mapping products. As you pointed out, there are some existing burned-area products, such as the burned area products based on Landsat (30m) and Sentinel (20m). And these products were generated based on the reflectance information and the strategy of land cover classification, instead of thermal information. These products indeed provide useful reference for study fire impacts on ecosystems, another critical domain of vegetation fires in the field of Earth system science. However, despite the high spatial resolution, these burned areas products had a much lower temporal resolution (the temporal resolution for Landsat and Sentinel fire products is larger than 16 days and 10 days respectively). In this case, these products cannot support real-time monitoring of active fires. As a comparison, MODIS and FY-3D active fire products were generated based on the extracted thermal information, which is different from the principle for producing burned-area maps. MODIS and FY-3D fire products, despite a coarse spatial resolution, present a high temporal resolution (daily), which can effectively monitor the occurrence and spread of large-scale fires (e.g. the Australia national fires or the Amazon fires). Thanks so much for this comment. We would make the difference between two types of fire products clearly in the revised manuscript.**

When the authors mentioned "continuous degradation" of mainstream global fire products. I don't know what the authors are referring to. I don't see this degradation. ESA is actively working on 20m-resolution burned area mapping which reveals much higher burned area in Africa than moderate resolution data.

- Is there degradation in active fire mapping by MODIS? I am not sure on this though. But at least MODIS active fire product provides a long term data covering about 20 years and there no other better data so far (prior to the publishing the current data in the paper).

**R: This is a good question and sorry that we did not make this clearer in the previous manuscript. For ESA's Sentinel burned-area products, which have been available for around 5 years, we also believed its quality and duration, as the sensors have been running for a relatively short period.**

**Actually, we mentioned that "continuous degradation" of mainstream global fire products mainly referred to the potentially growing system errors caused by long-term service. We are sorry that we did not use the expression more properly and might cause the confusion. Actually, we are not questioning the reliability of current MODIS active fire products. As explained above, we highly appreciated the high quality of MODIS fire products and their 20 years' time-series, which is not replaceable by any other products, including our FY-3D fire products (the time series was less than 5 years). However, a diversity of references (added in the revised manuscript) suggested that long-term running increases the aging of sensors and gradually increase the risk of system errors in the products. For instance, Landsat 7 products in the last two year**

presented more system errors than previous years. Therefore, in the revised manuscript, we revised the previous expression and suggested that the gradual ageing of sensors will, if not already, cause the future degradation of MODIS fire products. In this case, if scholars would like to continuously make full use of the long-term series of MODIS fire product, when it degrades significantly or even stops services in the future, a fire product with good reliability and similar characteristics is urgently needed. And the good accuracy and high consistence with MODIS fire products (here we regarded the reference data set with by far the best reliability), FY-3D fire products could be an ideal alternative and continuity of MODIS global fire products.

Thanks again for your comments. We have revised the manuscript accordingly to avoid unnecessary confusions.

- The paper lacks serious ground validation of the product. The only serious validation is shown in Fig. 5, but the paper only says the ground truth data are derived from visual interpretation (line 371). More information is absolutely needed on ground truth data to demonstrate that such data can be trusted and provide ground truth.

R: Again, thanks so much for this valuable comment. As explained above, a unique challenge for active fire products is how to evaluate the accuracy of the real-time monitoring of occurring fires (instead of the burned areas). For burned areas products, it is much easier to conduct field survey to gain ground truth data for verification, as the burned areas would remain unchanged for days or weeks. As a comparison, active fire products refers to the fires identified during burning process. When the fire stopped, the ground observation truth cannot be acquired, as the verification for active fire products lies in the capability of capturing occurring fires according to instant thermal information. Since active fires may last a very short period in a spot, depending on the underlying surfaces, it is not feasibly to stay there and wait for the potential occurrence of fires in given spots. A possible solution is to conduct wildfire experiments, which could only be carried out in a small region. However, even small-scale burning fire experiments are strictly restricted. Therefore, the acquisition of ground validation for FY-3D and other global fire products, including MODIS fire products was extremely difficult and thus previous relevant studies rarely considered the collection and use of field-collected ground truth reference. To verify the reliability of active fire products, there are three possible ways. First and the most frequent employed approach, visual interpretation based on expert knowledge. Second, treat one fire product as reference and compared this reference product with the target product, which is also frequently employed. Third, verification based on ground truth, which is most accurate, yet difficult and rarely employed by previous studies. In the previous manuscript, we employed visual check for quantifying the quality of FY-3D fire products globally. Meanwhile, we consider MODIS fire products as the reference data with good accuracy, and conducted the consistence analysis. In the revised manuscript, despite the extreme difficulty, we applied for some very rare ground observation reference data from a large field firing experiment conducted by China Meteorology Bureau, which further proved the

suitability of FY-3D fire products in China. This field fire-burning experiments were conducted in several provinces in China and a large body of drones were employed to immediately detect the active fires and reported to the team. In this case, ground truth data were collected. This data set was so far the very rare and the most official ground truth data for verifying active fire products. This is the first time, to our best knowledge, for such data employed in relevant research.

So in general, the global reliability of FY-3D fire products were proved by visual interpretation and consistence check with MODIS data while the suitability of FY-3D fire products in China was proved by the newly employed ground truth data.

Thanks again for your comments. By explaining this and adding the ground truth data, this manuscript has been largely improved.

- How about omission and commission errors? These are basic metrics used in satellite product evaluation. But such information is absent in the current paper.

R: This is a good point and thanks for pointing it out. To have a global coverage and high temporal resolution, the spatial resolution of active fire products, FY-3D and MODIS fire products, was relative coarse, 1km. On the other hand, the identification of fires requires the sensitivity of sensors. Therefore, for small fires, which simply took up a limited proportion in a cell. And there is a minimum size of fires which required to be recognized by MODIS or FY-3D sensors (e.g. 100 $m^2$). And by setting the minimum threshold of temperature difference (8K-10K), both MODIS and FY-3D regarded the omitted fires were attributed to the system limitations (small fires failed to meet the minimum firing area and cannot be identified in a 1km cell), not caused by algorithm limitations. Therefore, MODIS and FY-3D global fire products did not mainly consider the evaluation of omitted fires, which can only be identified through complicated ground experiments, instead of visual interpretation. Meanwhile, the commission errors caused by edge of water bodies and clouds and bare ground could be identified through visual interpretation. In this regard, at the global scale, the accuracy assessment of MODIS and FY-3D fire products, without ground observation truth, could not consider omission errors and the errors were mainly commission errors. So in the first part of this manuscript (the accuracy assessment of FY-3D fire products at the global scale), we simply employed the identification error, which were mainly commission errors.

Meanwhile, at the regional-scale, based on the burning-fire experiments, we could employ drone to report omitted fires. So in the third part, based on the ground observation truth, we further classified the identification errors as omission and commission errors for both MODIS and FY-3D products.

Thanks again for this constructive comment.

- In my first general comment, I elaborate on the usefulness of burned area product in the field of Earth system science. For active fire product, as is also stressed in the paper by the authors, the major usefulness is fire monitoring. But this will need fast response and accuracy. This comes to the major evaluations standard used in the paper when they compare with MODIS data. I think a 0.03 °threshold error for the monitoring purpose is quite of high tolerance. This means the error could be permitted 3 times as much as the spatial resolution of the product? Is this too big?

**R: This is a good point and thanks so much for the comment. We are sorry that this is a typo, which should be 0.02 °. Yes, after projection and resampling, the theoretically optimal resolution was 0.01 °. However, there were some bias for satellite positioning and most fires were sub-pixel information, the error tolerance is required. If the resolution was set as 0.01 °, the positioning error would cause the same fire pixel recognized as two different fire pixels in FY-3D and MODIS fire products. Meanwhile, although very small, there was a temporal difference between MODIS and FY-3D fire products (less than 1 hour), which also had an influence on the matching between two products. Thirdly, since the actual fire existed in sub-cells (which means it covers an area much smaller than a cell), this should also be considered in the data matching. For all these factors, we considered a buffer for a pixel and set the range of matching as 0.02 °.**

**Thanks again for this important comment.**

- In the comparison with MODIS data, what do you do with pixels which are identified as active fire but not shown in your data? I don't understand how 'matching' and 'mismatching' can cover the concept of commission and omission errors.

**R: Thanks so much for this comment. As explained above, at the global scale, without ground observation truth, both FY-3D and MODIS mainly considered the commission errors. Meanwhile, for this research, without the global truth data, we mainly treated the consistence with the MODIS data as one major index to explain FY-3D's consistence with MODIS fire products and the possibility of serving as its alternative products globally. Therefore, we simply considering the "matching" (consistence) between FY-3D and MODIS fire products at the global scale. Actually, due to a similar identification strategy and smaller minimum threshold for temperature difference, FY-3D rarely omitted active fires that could be identified by MODIS.**

**With ground truth data in China, we further considered the "omission" and "commission" errors for both MODIS and FY-3D data.**

**Thanks again for your comments.**

- I suggest authors could consider a target remote sensing journal, where more specialized reviewers are more likely available and can provide more relevant comments to help improve the quality of the work.

**R: Thanks for your comment. Actually, ESSD, as one of the best DATA related journals, included many remote sensing related papers and many specialized experts contributed significantly during the open discussion period. In addition to the expertise of another reviewer and the handling editor, clearly you are also an expert in remote sensing. Although you may not be familiar with some special characteristics of active fire products, your constructive comments were very helpful and the manuscript has been improved significantly accordingly. So we hope the largely revised manuscript can address your concerns and presented a good quality for ESSD, based on which global users could better understand and employ this data sources.**

**Thanks again for all these constructive comments and please feel free if additional revisions are required.**

- The author should work to improve readability. So far the english and arrangement are fine, but to follow the paper needs quite a lot effort.

**R: Thanks so much for your comment. We have carefully revised the manuscript according to the comments from you and other reviewer, and polished the English with all authors and English editors.**

Nonetheless, I also have some minor comments, which I hope can help:

Line 13: I don't see what the author mean here. There are uncertainties by different products. But I don't see they are in 'continuous degradation'. The authors have to provide evidence when making such a conclusion.

**R: We are sorry that we did not make this part clearer and caused some confusion. We acknowledged that MODIS fire products presented a generally good accuracy and a more than 20 years' time series, which is no doubt the best source for a diversity of long-term research. However, although it has been the best fire products for twenty years, the gradual ageing of sensors will, if not already, cause the future degradation of MODIS fire products. In this case, if scholars would like to continuously make full use of the long-term series of MODIS fire product, when it degrades significantly or even stops services in the future, a fire product with good reliability and similar characteristics is urgently needed. Against this background, we proposed our FY-3D global fire products as a promising alternatives and continuity for the existing MODIS fire products.**

**Thanks so much for this comment. We have revised the phrase here to avoid improper description and added relevant reference to prove that "long-term running could cause the degradation of sensors and possible increased system errors".**

Line 35: " millions of lost wildlife". pls check.

**R: Here we mean that massive wildlife could be killed in regional or national wild fires. For instance, millions of kangaroo and other animals were killed in the Australian national forest fires.**

Line 49-52: The citation of MODIS fire products of Giglio et al. 2003 is outdated. The most recent, as far as I know, is Giglio et al. 2018 (https://doi.org/10.1016/j.rse.2018.08.005). Also, I think the temporal resolution is daily. Pls check.

**R: Thanks so much for pointing this. We have revised it accordingly.**

Line 64: what is the sense of this 16 days temporal resolution ?

**R: Sorry that we did not make this clear. This means the 16 days' revisit cycle of Landsat-8.**

Line 70-72: citation needed.

**R: Thanks for this comment. Citations have been added.**

Line 83-84: Wang et al. 2012 evidence for decrease in reliability of MODIS fire products? What are the quantitative standards for 'high-quality'?

**R: Sorry that we did not expressed this properly in the previous manuscript. Actually, there were no global quantitative evidence for quality of MODIS data. We have added more reference to suggest that theoretically and qualitatively, the gradual ageing of sensors will, if not already, cause the future degradation of satellite products. For instance, the quality of Landsat 7 products degraded in the last two years. And our FY-3D fire products, with similar spatiotemporal resolution, good accuracy and high consistence with MODIS fire products, could serve as a safety code to prepare for the future degradation or stop-service of MODIS global fire products.**

**Thanks for this comment, we have carefully revised the phase in the introduction and discussion part to avoid unnecessary confusion.**

Line 115-118: what is the relevance for fire mapping?

**R: Thanks for this comment. We have revised this part to make it more concise.**

Line 241: should the identification of cloud pixel meet any one these criteria in Table 3?

**R: Yes, any one of these criteria can help us identify this pixel as cloud pixel. We added more explanation here to make it clearer.**

The brightness temperature seems referring to all channels of 20, 24 and 25. The authors should state specifically which of them they are referring to when using this in the article. E.g., line 250. Or at least clear definition is needed to indicate which bands the authors are referring to when talking about "background rightness temperature", and other terms.

**R: Thanks so much for pointing this out and in the revised manuscript, we had made it clearer. Actually, both middle-infrared and far-infrared bands, mainly middle-infrared bands were employed. Specifically, middle-infrared band provides abnormal thermal information while far-infrared bands provides surface radiation information, which prevent commission errors caused by reflectance information.**

Line 453: The authors confuse on the concept of 'precision' vs. 'accuracy'. see https://en.wikipedia.org/wiki/Accuracy_and_precision Accuracy and precision are two measures of observational error. Accuracy is how close or far off a given set of measurements (observations or readings) are to their true value, while precision is how close or dispersed the measurements are to each other. Here the practice could be best called a consistency test with MODIS data, because neither MODIS or FY-3D gives the true value.

**R: Thanks so much for this constructive comment. Yes, as explained in details, this research mainly focused on the accuracy assessment of FY-3D products globally based visual check and consistence check with MODIS products globally. Meanwhile, we added ground truth data to verify the suitability of FY-3D and MODIS fire products in China.**

**So you are right and we should use the words more carefully. In the revised manuscript, we have checked it accordingly.**

Line 100: FY-3 => FY-3D?

**R: Thanks for pointing this out. Corrected.**

Line 379-381: the text says that the threshold is 0.02 °but the equation says a threshold of 0.03 ? Which one is the correct? Also see line 427.

**R: We are sorry that this is a typo, which should be 0.02 ?**

Line 425: given the scope of the whole globe, visual checking of underlying surface areas to exclude water, ice, snow and bare land seems quite challenging. Are there a lot pixels ?

**R: Yes, you are right. For checking the consistence between FY-3D and MODIS fire products in 2019, our research team in China Meteorology Bureau examined 2237714 matched pairs of FY-3D and MODIS fire products.**

Line 423-425: if MYD14A1 has a daily temporal resolution, is this in contradiction with that data set with observation time less than 1 h is selected?

**R: Sorry that we did not make it clear. Less than 1h means the difference of local observation moment for FY-3D and Aqua was very close. As we know, the larger the temporal difference between the visiting moment of two sensors, the larger uncertainties involved. Fortunately, the less-than-1h difference between Terra and FY-3D is so far the smaller than other satellites, which made FY-3D fire products an ideal replacement and continuity of current MODIS products.**

Line 123: Product overview. I suggest the authors put a table specifying the main characteristics of the data, saving the readers' time to find them through the texts.

**R: Thanks so much for this comment. We have added a table to the revised manuscript.**